# Adaptive Thermal Image Velocimetry of spatial wind movement on landscapes using near target infrared cameras

Benjamin Schumacher[1], Marwan Katurji[1], Jiawei Zhang[1, 4], Peyman Zawar-Reza[1], Benjamin Adams[2], and Matthias Zeeman[3]

[1]School of Earth and Environment, University of Canterbury, Christchurch, New Zealand
[2]Department of Computer Science and Software Engineering, University of Canterbury, Christchurch, New Zealand
[3]Institute of Meteorology and Climate Research - Atmospheric Environmental Research, Karlsruhe Institute of Technology, Campus Alpin, Garmisch-Partenkirchen, Germany
[4]New Zealand Forest Research Institute (Scion), Christchurch, New Zealand

**Correspondence:** Benjamin Schumacher (benjamin.schumacher@pg.canterbury.ac.nz)

**Abstract.** Thermal Image Velocimetry (TIV) is a near-target remote sensing technique for estimating two- dimensional near-surface wind velocity based on spatiotemporal displacement of fluctuations in surface brightness temperature captured by an infrared camera. The addition of an automated parameterization and the combination of ensemble TIV results into one output made the method more suitable to changing meteorological conditions and less sensitive to noise stemming from the airborne sensor platform. Three field campaigns were carried out to evaluate the algorithm over turf, dry grass and wheat stubble. The derived velocities were validated with independently acquired observations from fine wire thermocouples and sonic anemometers. It was found that the TIV technique correctly derives atmospheric flow patterns close to the ground. Moreover, the modified method resolves wind speed statistics close to the surface at a higher resolution than the traditional measurement methods. Adaptive Thermal Image Velocimetry (A-TIV) is capable of providing contact-less spatial information about near-surface atmospheric motion and can help to be a useful tool in researching turbulent transport processes close to the ground.

## 1 Introduction

### 1.1 Atmospheric turbulence and its implications on surface temperature

Atmospheric boundary layer turbulence is a key driver for energy transport and dissipation between the surface and the atmosphere (Stull, 1988). Turbulence can be organized in coherent structures which are estimated to be responsible for at least 40% of the surface momentum and heat fluxes and largely contribute to transport processes within the atmospheric boundary layer (Lotfy et al., 2019; Barthlott et al., 2007; Litt et al., 2015). Atmospheric turbulence and the coherent structures are therefore responsible for surface temperature fluctuations especially during short time periods ($< 1$ min) when radiative input is relatively constant (Christen et al., 2012). This effect has lead to the study of surface atmospheric interactions using brightness temperature measurements. Paw-U et al. (1992) used an infrared thermometer to identify ramp structures in the surface temperature over a maize canopy. They found that the surface temperature ramps of these structures are smaller in magnitude than the air

temperature ramps but followed a similar pattern with a high correlation. Katul et al. (1998) linked surface temperature fluctuations over a forest clearing to the turbulent velocities measured near the surface and concluded that for cloud free conditions turbulent velocities can induce large brightness temperature perturbations ($> 2°C$).

Time Sequential Thermography (TST) is a methodology introduced by Hoyano et al. (1999) referring to ground-based thermal infrared cameras sampling at sub-minute intervals providing spatial and temporal information about surface brightness temperature changes. TST methods depicting the imprints of turbulent coherent structures have been used to draw conclusions about surface atmospheric interaction mechanisms (i.e. sweep and ejection mechanisms), the surface materials and their potential brightness temperature fluctuation and the shape and movement of turbulent coherent structures in the surface layer of
the atmospheric boundary layer (Hoyano et al., 1999; Garai and Kleissl, 2011; Christen et al., 2012; Garai et al., 2013).

Garai and Kleissl (2011) described the detection of coherent turbulent structures using their thermal footprints and concluded that TST can only provide information under certain conditions, i.e., low vegetation and a flat homogeneous surface due to vegetation movement and the preservation of heat in the vegetation canopy allowing for little brightness temperature fluctuations.

Infrared cameras provide advantages for spatial turbulence studies over using traditional approaches such as arrays of sonic anemometers (Inagaki and Kanda, 2010). Firstly thermal cameras are not invasive of the turbulent flow field. Secondly the measurement is instantaneous and spatial and therefore no interpolation of point-based measurements is needed. Spatial turbulence studies also utilized Particle Image Velocimetry (PIV) techniques which seed reflective particles in air visualizing the turbulent flow using image correlation techniques (Adrian et al., 2000; Hommema and Adrian, 2003). However, PIV implies
a number of limitations in a geophysical environment including small covered areas ($< 100~m^2$) due to the particle size, the seeding density and the intensity of the light pulse (Hommema and Adrian, 2003; Inagaki et al., 2013).

## 1.2  Thermal Image Velocimetry

Inspired by PIV techniques Inagaki et al. (2013) suggested a method called Thermal Image Velocimetry (TIV) for estimating advection velocities of thermal structures over artificial surface types like polystyrene boards and turf. A high correlation of
these velocities with near surface wind velocity measurements was reported. However, when moving from a static setup and artificial surface covers to an airborne (helicopter-based) acquisition of the thermal imagery over a forest, TIV could not be calculated due to image shaking (Inagaki, 2016). The removal of the image shaking and the accuracy assessment of different correlation techniques within the TIV process was described by Schumacher et al. (2019) showing that TIV from hovering Uncrewed Aerial Vehicles (UAVs) can be accomplished.
The TIV as presented by Inagaki et al. (2013) implies a number of limitations on the retrieval of the two-dimensional TIV velocity vectors which for instance require an experienced user to set a large number of user input parameters, as well as the algorithms capability to retrieve velocities only on artificial, smooth surface types with a low thermal conductivity, high emissivity and small heat capacity. Moreover, the retrieved TIV spatial thermal pattern displacement velocity has not yet been compared and evaluated against near-surface spatial wind velocity measurements.

In a recent study Alekseychik et al. (2021) employed PIV techniques to thermal imagery collected with a UAV to retrieve spatial TIV wind fields which were averaged over 80 seconds and compared to average measurements from a sonic anemometer. The study was focused on identifying coherent structures interacting with the surface using the brightness temperature perturbations and deriving statistics about size and shape of the interactions. However, the used techniques did not resolve instantaneous velocities of the coherent structures which is important for their evolution and shape as well as the interaction with the surface.

## 1.3 Thermal Imagery in remote sensing

Thermal imagery acquired from towers have been utilized for highly resolved land surface temperature in space and time for spatial heat flux calculations in surface energy balance models (Morrison et al., 2017; Garai et al., 2013). Other studies have utilized thermal imagery acquired with moving UAVs as land surface temperature substituting the comparatively low spatiotemporal resolution of satellite acquisitions to estimate evapotranspiration in surface energy balance models (Brenner et al., 2017, 2018; Simpson et al., 2021; Bastiaanssen et al., 1998). In both types of studies wind velocity is commonly represented by single point anemometer measurements which can be considered a weakness of the surface energy balance models (Waters, 2002). Neither TST nor TIV have been applied to retrieve spatial wind velocities for the benefit of theses estimations.

Due to the limitation of TIV to artificial surface types and the limitation of use from stable oblique camera towers, TIV has not yet been adapted into surface energy balance estimations. Because of these current limitations neither TST nor TIV have provided a landscape scale spatial velocimetry estimate over natural surfaces. Nonetheless, a spatial two-dimensional near surface wind velocity measurement in a landscape scale would be beneficial across scientific disciplines. For example, in atmospheric science this measurement offers the opportunity for desired validation or calibration of numerical weather models which are currently commonly validated with *in-situ* point measurements (Sagaut and Deck, 2009; Giordano et al., 2013). In agricultural and environmental science two-dimensional near surface wind velocity would be valuable for estimations of vital environmental parameters such as evapotranspiration, water stress of plants or energy fluxes (Pozníková et al., 2018; Morrison et al., 2017).

In this study we validate the new development called Adaptive Thermal Image Velocimetry (A-TIV) which performs a Hilbert Huang Transform (HHT) analysis of the brightness temperature data before the calculation of the A-TIV and then uses multiple surface brightness temperature perturbation filter sizes to cover multiple scales of temperature perturbations. Subsequently, we pursue the validation with spatially distributed high-frequency fine wire thermocouple measurements as well as using sonic anemometer data which are compared to the A-TIV measurements. Through the spatio-temporal high frequency thermal pattern displacement velocities covered by the developed A-TIV and their comparison to *in-situ* measurements, new insights into the interaction of turbulent coherent structures with the Earth's surface become possible. With three different surface types and canopy heights this study also tests the limitations of the A-TIV algorithm. The research objectives are:

1. To prove that UAV acquired and stabilized surface brightness temperature over artificial turf and non-artificial grass surface is reflecting the near-surface atmospheric flow patterns.

2. To improve the usability of the TIV algorithm and eliminate necessary user-input parameters by analyzing the input thermal imagery before the calculation. This process included also building an open-source algorithm with automated components, making A-TIV and TIV available to non-experts.

3. To perform a validation with independent measurements to assess the accuracy of the A-TIV algorithm over artificial turf and non-artificial grass surface cover.

4. To test the limitations of the A-TIV algorithm using a third surface cover: wheat stubble. The wheat stubble experiment is dedicated to analyse the limitations of the proposed A-TIV algorithm in terms of vegetation height.

## 2 Methodology

### 2.1 Thermal Image Velocimetry Algorithm

Thermal Image Velocimetry is a method to spatially estimate thermal pattern velocity through the tracking of surface brightness temperature fluctuations measured by an infrared camera at a high frequency (>1 Hz). The success of the TIV algorithm depends on a set of user inputs especially the correlation window size, the search area size, the correlation time interval, and the temporal running filter size for the perturbation calculation (see Equation 1). As illustrated in Figure 1 - B) and C), the correlation window size defines how many vectors are calculated in the image, the search area size defines the density of the vectors, the time interval is decisive for the vector length and for the number of error vectors calculated, and the running filter size determines the noise level of the perturbation calculation (Inagaki et al., 2013; Schumacher et al., 2019).

TIV used previously a correlation technique presented by Kaga et al. (1992) called the greyscale correlation technique which uses simple pixel by pixel subtraction to obtain a correlation value (Inagaki et al., 2013). The A-TIV is usually calculated using the same technique with a correlation window size of 16 x 16 pixels and a search area size of 32 x 32 pixels. These settings were previously investigated as the most accurate (Schumacher et al., 2019).

$$T'_s = T_s - \overline{T_{sF}} \tag{1}$$

Equation 1: Calculation of perturbation for the data cube. $T'_s$ is the resulting perturbation of one pixel, $T_s$ is the measured brightness temperature of the pixel and $\overline{T_{sF}}$ is the temporal mean of the pixel dependent on the filter size F (5, 10, 20, or 30 seconds). See Figure 1 B) for context.

### 2.2 Adaptive Thermal Image Velocimetry

The A-TIV algorithm is based on the same principle as the TIV algorithm with two new modifications (Figure 1 – A) red box and D) ), which allows an automatic adaptation of the algorithm to its input brightness temperature imagery and therefore the

underlying environmental conditions. In previous investigations, the TIV algorithm showed a large sensitivity to the correlation time interval setting, delivering erroneous vectors when the setting was too short (see Section 3.2.1). Therefore, as a first development A-TIV takes advantage of the Hilbert-Huang transform (HHT) identifying the instantaneous frequencies of a given signal (Huang et al., 1998). In the case of A-TIV the first step of the HHT is to calculate the Ensemble Empirical Mode

Decomposition (EEMD) which determines the Intrinsic Mode Functions (IMFs) of the time series from eleven randomly selected pixels from the input thermal imagery (Wu and Huang, 2009). The second step is to calculate the Hilbert Transform of the first IMF of each selected pixel defining the highest instantaneous frequencies of the input thermal imagery. The average of the eleven calculated instantaneous frequencies prescribe the correlation time interval setting for the A-TIV algorithm. This allows to change the interval setting based on variable experimental input data.

As a second development, A-TIV makes use of sequenced time filters in the perturbation calculation (see Equation 1). The temporal filter size F in $\overline{T_{sF}}$ of the TIV algorithm is varied to cover multiple magnitudes of perturbations in the brightness temperature data set (see Figure 2). When multiple perturbation filters are calculated on the stabilized brightness temperature data set the resulting perturbation from the longest filter size F represents the largest scale of motion. One TIV sequence is calculated on each of these 3-dimensional perturbation data cubes ($x$, $y$, time; see Figure 1 ). All resulting velocity data cubes

are subsequently assimilated from longest to shortest perturbation filter size using a weighted average with the filter size as weight (in this study: 30 seconds, 20 seconds, 10 seconds and 5 seconds or sixfold, fourfold, twofold and onefold). This leads to a velocity field output which contains more information from the longest perturbation filter sizes. An increasing noise level of the camera is present with decreasing perturbation filter size. Therefore the weighted average is based on the perturbation filter size which reduces the importance of the TIV output calculated from the smallest perturbation filter. This allows the

A-TIV algorithm to include all velocity fields with reduced noise associated disturbances. The calculation of the A-TIV also removes outliers of the output wind velocities by replacing values that are outside of 2 standard deviations with the mean value within a 3-by-3 window (see Figure 1 C) ).

    According to Inagaki et al. (2013) perturbation filter sizes ($\overline{T_{sF}}$) from 1 second to 30 seconds show a high correlation with near surface measured wind velocity when calculating TIV. Therefore, the default setting is four different running filter sizes:

30 seconds, 20 seconds, 10 seconds, and 5 seconds.

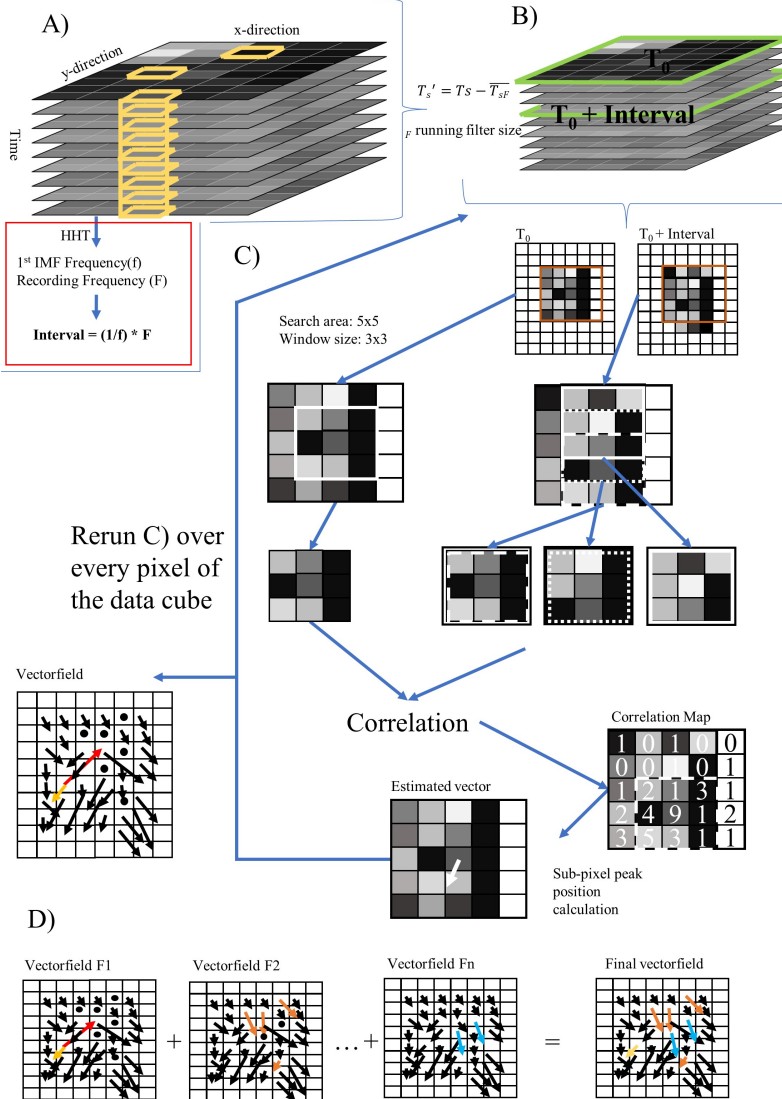

**Figure 1. Schematic of the A-TIV algorithm** -

A) Randomly selected stabilized brightness temperature pixels (marked orange) are used with HHT to find the highest frequency signal component of the data set which defines the interval setting.

B) The perturbation calculation with different predefined running filter sizes (F) is done to provide the data for computation of the TIV core algorithm. From the perturbation data set two images with the time increment of the interval setting are extracted and passed to C).

C) Core TIV algorithm with an example search area size of 5 x 5 pixel and a correlation window size of 3 x 3 pixel. Error vectors are adjusted using the standard deviation of the calculated vectors in the direct adjacent pixels.

D) Process from C) is computed over all predefined filter sizes of B) to calculate multiple vector fields are available. Finally, all vector fields are merged using weighted averaging.

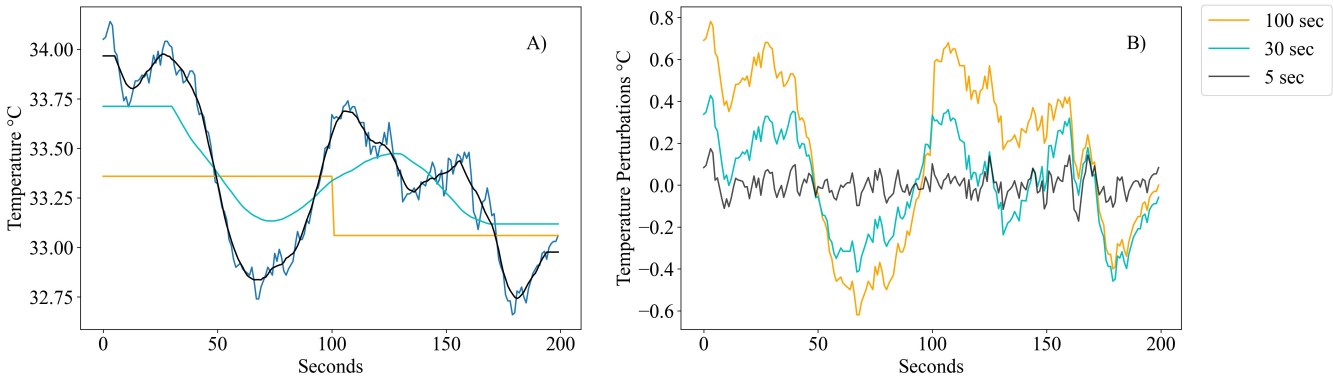

**Figure 2. Sample Pixel: A) Temperature and B) Temperature perturbations** - Example showing impact of different running filter sizes to calculate the perturbations. A) Temperature measurement from a sample pixel of the IR Camera. Dark blue is the original temperature, and the smoothed lines are the running mean used for the perturbation calculation (see legend for the length). The corresponding perturbation scales are shown in B). The mean of 100 seconds only allows large perturbation influences to be shown while the smaller perturbation window of 5 seconds accounts for small scale changes.

## 2.3 Data

Three separate field experiments in the Canterbury Region of New Zealand were carried out during the Australasian summer in January and March 2019 and in January 2020. The two Time seqUential theRmal inFrared Turbulence (TURF-T) experiments were equipped with sonic anemometers, fine wire thermocouples, and an infrared camera on a UAV (Figure 3). The first experiment TURF-T1 took place at a hockey turf field and the second experiment TURF-T2 was held at a mixed surface with turf and grass. The third experiment included an infrared camera on a UAV acquiring footage over a dry wheat stubble field with a weather station in the proximity of 100 metres collecting 1-minute average wind speed. This third experiment was designed to test the algorithm when used over a higher vegetation canopy due to missing high-frequency measurements in a qualitative analysis. The weather station data was used to monitor the atmospheric conditions during the experiment day and put the A-TIV output from this experiment into context to the TURF-T1 experiment and TURF-T2 experiment.

The wheat stubble experiment took place after the harvest of the crops when the individual stalk heights were cut to 18 – 20 cm and left standing upright in the field. Compared to the spikes of the wheat plant the stalks are a very stable part of the plant and it was expected that even with higher wind gusts the stalks will not create any motion effects also called "honamis" interfering with the camera measurement (Finnigan, 2010). Table 1 provides a summary of the three experiments and Table 2 provides a summary of the available instruments.

The surface types were picked in accordance with other field experiments, which suggest that the application of the TIV method is more suitable for dry and thermally responsive surfaces with a high thermal admittance (Inagaki et al., 2013; Christen et al., 2012). For the acquisition of the infrared video the Optris PI 450, a lightweight, passively cooled camera with the capability to be attached and stabilized to a UAV using a 3-axis dynamically responsive camera mount, was used (Figure 3, Image 3). The camera measures a spectral range of 8 – 14 μm, in a resolution of 382 by 288 pixels, with a system accuracy

of $\pm 2°C$ or 2% of the measured temperature and a thermal sensitivity of 40 mK (Optris, 2020). Due to the passive cooling mechanism of the camera, its sensor is reset for one second in a frequency of ten seconds. The camera lens used for all experiments is a wide angle lens with a field of view of 80°x 54°an a fixed aperture of f/6. The acquisition frequency was set to 80 Hz, the output file format is a radiometric video file.

In addition to the brightness temperature measurements, near-surface temperature and wind velocity measurements were made to compare the brightness temperature measurements and A-TIV wind speeds to the *in-situ* meteorological measurements. TURF-T1 was equipped with one sonic anemometer and TURF-T2 was equipped with two sonic anemometers, one in the grass field and one in the turf (Figure 3). All anemometers were mounted at 1.5 m above ground level, sampled at 20 Hz, and were placed in the field of view of the camera. Additionally, the TURF experiments were equipped with an array of

fine wire thermocouples (TC - 0.0254 mm bead diameter) mounted approximately 1.5 cm above the ground, measuring air temperature with a sampling speed at 20 Hz (Figure 3, Image 2). The 12 sensors in the TC array were distributed in a square with three sensors per edge (eight sensors total) and four sensors aligned in the bisector of the square (Figure 3) resulting in an interdistance of 1.5 m. This distribution ensured a proper lag correlation methodology to retrieve near-surface wind direction and wind velocity.

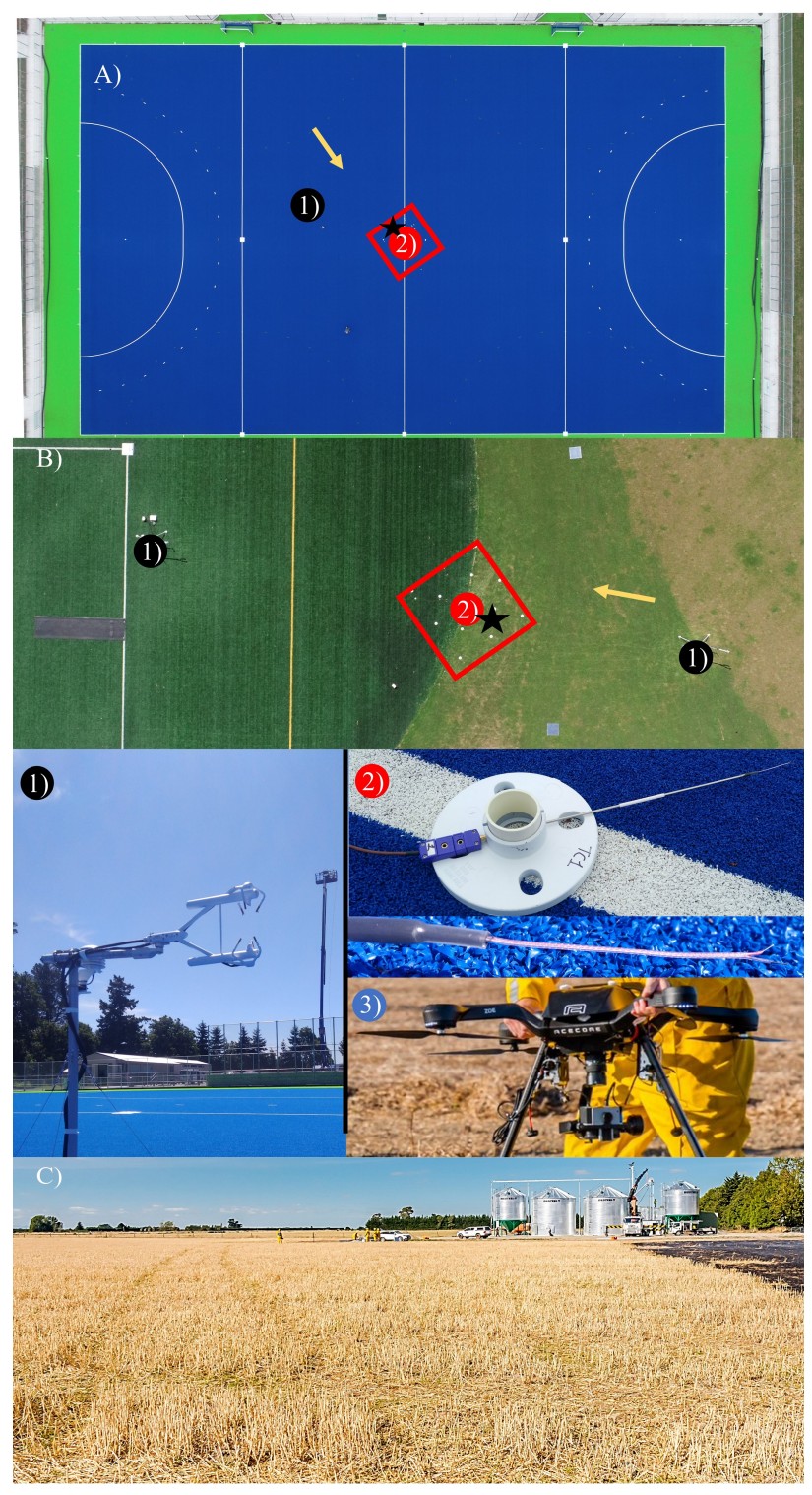

**Figure 3. Experiment sites and instrumentation** - TURF-T1, TURF-T2 and wheat stubble experiment sites: A – blue hockey turf, B – green softball turf with surrounding grass and C - a harvested wheat field with stubble. In A) and B) the positions of sonic anemometer units (black dots, Image 1 – closeup below) and the thermocouple array (red square with red dots, Image 2 – closeup below) are marked. The black star shows the position of the base thermocouple upwind which was used for close surface wind velocity estimation (see section 2.7). The mean wind direction is emphasized by a yellow arrow. Image 3) Shows the used quadcopter with a 3-axis inertial stabilizer and the Optris camera mounted.

## 2.3.1  Meteorological conditions

Experiments were carried out when the Canterbury Region was under the influence of stagnant anti-cyclonic (high pressure) synoptic with weak pressure gradients resulting in near-surface wind speeds in the ranges of 2 – 5 m/s, however, we expect differences in the characteristics of the boundary layer development (and hence the turbulence field) for each day. This will be due to the variation in cloud cover and the fact that the characteristics of the thermally generated flows (such as sea-breezes) will be different for each day. The conditions on the 7th of March 2019 were additionally influenced by a low pressure system to the southwest of New Zealand which caused higher wind speed ranges during this day. The TURF-T1 experiment day was characterized by cloudy conditions in the morning progressing to clear sky conditions in the afternoon with an average temperature of 18.3 °C. This morning cloud cover did not interfere with the temperature during the TURF-T1 experiment. TURF-T2 was carried out with no cloud cover and an average air temperature of 20.6 °C. The wheat stubble experiment was carried out during a period with scattered high clouds and an average air temperature of 22.2 °C. For more detailed descriptions on the meteorological conditions during the experiments see Table 1. For an overview of the used instrumentation please refer to Table 2.

**Table 1.** Overview of the Experiments

| Experiment | Time | Mean WS | Mean WD | UAV height | Location | Surface Type + Dimension | A-TIV Inverval |
|---|---|---|---|---|---|---|---|
| TURF-T1 | 12-01-2019 16:21 – 16:31 | 2.6 m/s | north-easterly | 70 m | Rangiora Sportsground | Dry hockey turf 91.4 m x 55 m | 1.5 s |
| TURF-T2 | 17-01-2020 14:25 – 14:38 | 5.8 m/s | north-easterly | 35 m | Rolleston Sportsground | Mixed grass, turf 3 - 5 cm grass height 60 m x 30 m | 2 s |
| Wheat Stubble | 07-03-2019 17:34 – 17:39 | 8.4 m/s | northerly | 70 m | Darfield | Wheat stubble 18 - 20 cm stalk height 90 m x 52.5 m | 4 s |

**Table 2.** Overview of the Instrumentation

| Experiment | Wind velocity | Thermal Imagery | Thermocouples | Logging frequency |
|---|---|---|---|---|
| TURF-T1 | 1 x sonic anemometer (EC100) | Optris Pi450 | 16 Thermocouples Type E | 20 Hz |
| TURF-T2 | 2 x sonic anemometer (EC100) | Optris Pi450 | 16 Thermocouples Type E | 20 Hz |
| Wheat Stubble | 1 x automated weather station | Optris Pi450 | None | 1 minute |

## 2.4 Thermal Imagery Stabilization Process

The unstable brightness temperature collected with a UAV based system is stabilized using Blender, a 3D video animation software package (Cardona, 2006; Ramos et al., 2011; Blender Online Community, 2019). A detailed description of the process is available in Schumacher et al. (2019) which can be summarized as follows: To read the collected brightness temperature video into the Blender the data is transferred to RGB colour value images. The Blender software is employed to track the camera movements using the manual camera tracking feature and the low emissivity targets which display as cold spots in the imagery. Then the stabilization process was computed with a nearest neighbor interpolation on the RGB colour images using low emissivity targets in the field of view of the camera. Subsequently a random forest machine learning algorithm is trained on the unstable brightness temperature - colour value image pairs. The random forest algorithm in this case works as a colour value – temperature model. Finally, the stable temperature images were predicted from the stabilized colour images with the random forest algorithm. The effects of the stabilization for the TURF-T1 experiment are shown with the standard deviation of the images over time in Figure 4. Before using the stabilized brightness temperature, it should be subsampled using averaging from the original acquisition frequency to a suitable noiseless frequency. For example, the TURF-T1 experiment data was subsampled from 80 Hz to 2 Hz. The stabilized and subsampled video can be registered with a geographic coordinate system.

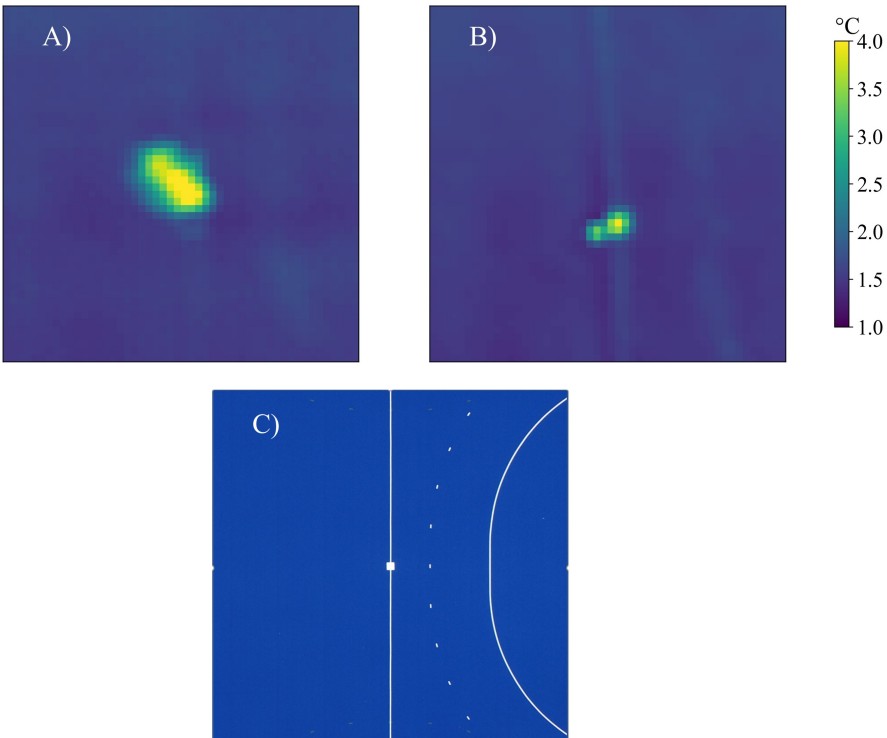

**Figure 4. Standard deviation over time for A) unstable and B) stabilized thermal imagery** - Impact of stabilization algorithm on the brightness temperature (Tb) standard deviation over time zoomed to a low emissivity target of the TURF-T1 experiment. A) shows the unstable Tb standard deviation, B) shows the stabilized Tb standard deviation. C) shows RGB imagery with the blue turf and the low emissivity target in the centre. The peaks of the standard deviation of A) and B) are shifted due to the working mechanism of the software stabilization. Any subsequent frame in the stabilization will be matched with the base frame (first frame) and the standard deviation peak hence is shifted towards this first frame.

## 2.5    Thermocouple Analysis

The Thermocouple (TC) array allowed us to estimate wind speed and wind direction very close to the surface, by estimating the lag cross-correlation on overlapping subsets (10 s) of TC time series data (20 Hz). The resulting 2 Hz TC-based advection wind velocity is calculated based on one TC measurement (base TC, black star in Figure 3) lagged over ten seconds and cross-correlated with all the surrounding TCs. The base TC was defined as one of the inner four TCs which was placed upwind of the other thermocouples. The TC-based advection wind speed was used in the accuracy assessment of the A-TIV algorithm.

The time lag and the distance of the TC with the highest correlation coefficient determined the wind velocity, and the position relative to the base TC determined the wind direction. Because of the overlapping ten second windows in the time series, the resulting frequency of wind speed and wind directions measurements were 2 Hz. Due to maximal and minimal lag-correlation times the minimal resolved wind velocity is limited to 0.25 m/s.

To evaluate in a first step the brightness temperature data captured by the infrared camera with the TC derived air temperature, the same methodology was applied to a "virtual" array taken from the brightness temperature perturbations which was sub-sampled using mean-sampling to a sampling rate of 20 Hz. The lag cross-correlation was calculated on this sampling rate resulting with a 2 Hz wind direction and wind speed measurement identical to the physical array calculation. For the TURF-T1 experiment the centre of this "virtual" array was located about 9 m upwind from the centre of physical TC array. The retrieved wind speed and wind direction were compared directly to the results from the physical array using a basic statistical t-test analysis and a histogram comparison. The same method was applied to the TURF-T2 experiment in which the physical array was distributed across both surfaces; hence, the "virtual" array was tested in 3 different positions: one in the grass field 9 m upwind of the physical array, one in the turf field 9 m downwind of the physical array, and one 1 m next to the physical array to resolve the same surface types for each TC.

## 2.6 A-TIV Evaluation

The sonic anemometer data from the three experiments were used to calculate the mean wind speed over the experimental period to characterize the different meteorological conditions during the experiments (Table 1). The evaluation of the A-TIV-based thermal pattern velocity involved a statistical analysis with a comparison to the statistics of the TC and EC measurements. For the comparison the A-TIV speeds were averaged over an area of 15 m by 15 m to ensure four homogeneous fields of turf (TURF-T1 and TURF-T2), grass (TURF-T2) and wheat stubble. In TURF-T2 the calculation of the A-TIV was separated for both surface types to be able to compare the results from the artificial to natural surface cover. The statistical analysis in-cluded an analysis of the probability density functions and a t-test comparison of the average of the thermal pattern speed from both TURF experiments to the TC measurements. Additionally, the algorithm performance from the wheat stubble experiment was analysed comparing the area average A-TIV speed histograms of all experiments. For all comparisons histograms and probability density functions are presented rather than instantaneous cross-correlations because of the difference in measure-ment approaches meaning that A-TIV is resolving a near-surface spatial velocity whereas the sonic anemometers represent a point measurement of a spatial footprint and the TC array measurement are representative of the immediate area where they are mounted. Therefore, the histograms and probability density functions provide a better overview of the similarities and differences than the time series from the measurements.

## 3 Results

### 3.1 Stabilized Brightness Temperature Evaluation

The TC array data was used to estimate wind speed and wind velocity approximately 1.5 cm above the ground using cross-correlation and the distance of the TCs (see Section 2.7). However, in a first analysis the brightness temperature (Tb) from the camera was used to create a virtual TC array to check if the data is depicting a similar measurement as the physical TC array. Figure 5 shows the histograms of the estimated wind speed and wind direction for TURF-T1 and TURF-T2 over grass and

turf. In TURF-T1 the calculated wind speed and wind direction of the virtual TC array did not differ significantly from the physical TC array (p-value > 0.95). When the virtual TC array was placed in either the grass or the turf surface in TURF-T2 no

similarity could be determined. However, when the virtual TC array was placed next to the physical array and the surface types of the virtual and the physical TCs matched a weak significant difference in wind speed distribution was estimated (p-value > 0.9).

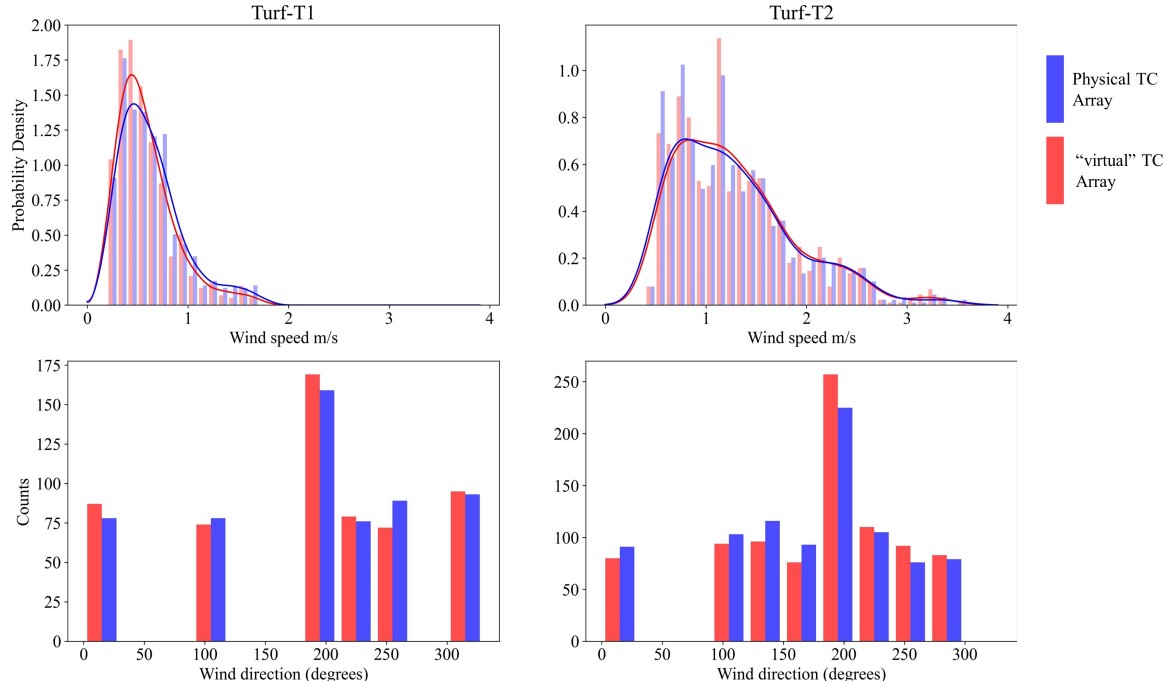

**Figure 5. TURF-T1 and TURF-T2 brightness temperature accuracy assessment** - Comparison of wind speed (m/s - top row) and wind direction (degrees - bottom row) of the physical TC array (red) to the virtual TC array (blue) for TURF-T1 and TURF-T2. It is evident that the lag correlation using the virtual arrays is depicting similar wind speed and direction as the physical TC array which measures air temperature approximately 1.5 cm above the ground in both experiments.

## 3.2   A-TIV Evaluation

### 3.2.1   TIV sensitivity to user input

The stabilization of UAV thermal imagery is the foundation for a successful TIV, however the A-TIV incorporates two steps to reduce error vectors (see Section 2.2) and create a more detailed output over non-artificial surface types. To show the advantages of the first step an example frame of the TURF-T1 TIV is displayed with the correlation time interval setting informed by the HHT and compared to one with a user-defined correlation time interval setting (see Figure 6). The TIV with the HHT as correlation time interval setting mechanism is removing error vectors and showing a larger spectrum of velocities. An error

vector in Figure 6 is defined as a vector which implies local, unrealistic measured A-TIV speeds (> 6 m/s) whereas during the experiment day the mean wind speed was measured at 2.6 m/s.

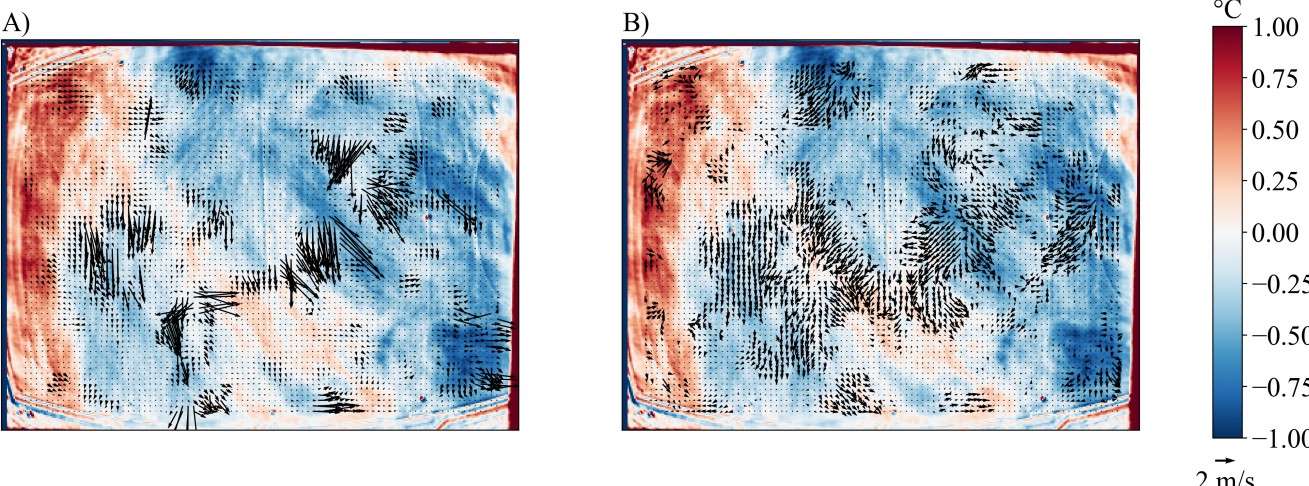

**Figure 6. Error vector removal due to correct correlation time interval setting** - A) shows a TURF-T1 TIV with a correlation time interval of 0.5 seconds. B) shows the same TIV with an interval setting of 1.5 seconds which was calculated using the HHT. The large error vectors mask the display of the small scale vectors within A).

The second step of A-TIV is the incorporation of the four perturbation calculations resulting in one TIV each which are subsequently assimilated using a weighted average (see section 2.2). Figure 7 shows an example frame of TURF-T1 with each of the four perturbation filters and the corresponding TIV with the HHT as interval setting mechanism. The product A-TIV for

this frame is the weighted average of these four TIVs.

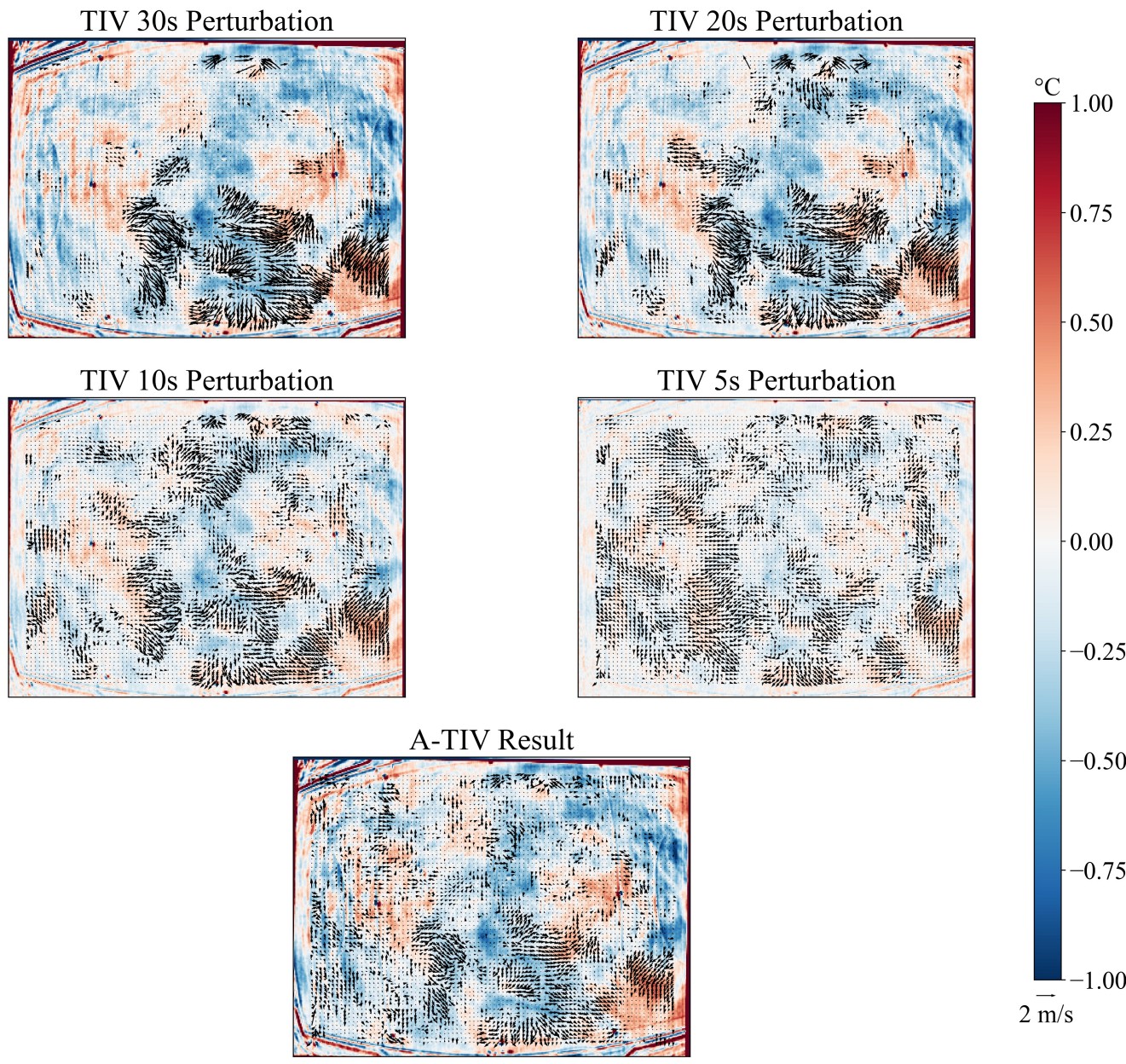

**Figure 7. TIV from four different temporal running filter sizes** - Extraction of TIV vectors from the four perturbation temporal running filter sizes 30 seconds - 5 seconds. It is evident that with decreasing temporal running filter size more vectors are calculated.

### 3.2.2 Differences of A-TIV and TIV

A-TIV adds velocities to certain areas where the TIV does not estimate velocities (Figure 6). The difference of A-TIV speed and TIV speed is displayed in Figure 8. A positive difference indicates that the A-TIV speed is higher than the corresponding

TIV speed of a certain perturbation filter size. Through the weighted averaging the A-TIV gains with each new layer new information until the lowest perturbation filter adds the least amount of information due to its weight in the weighted average.

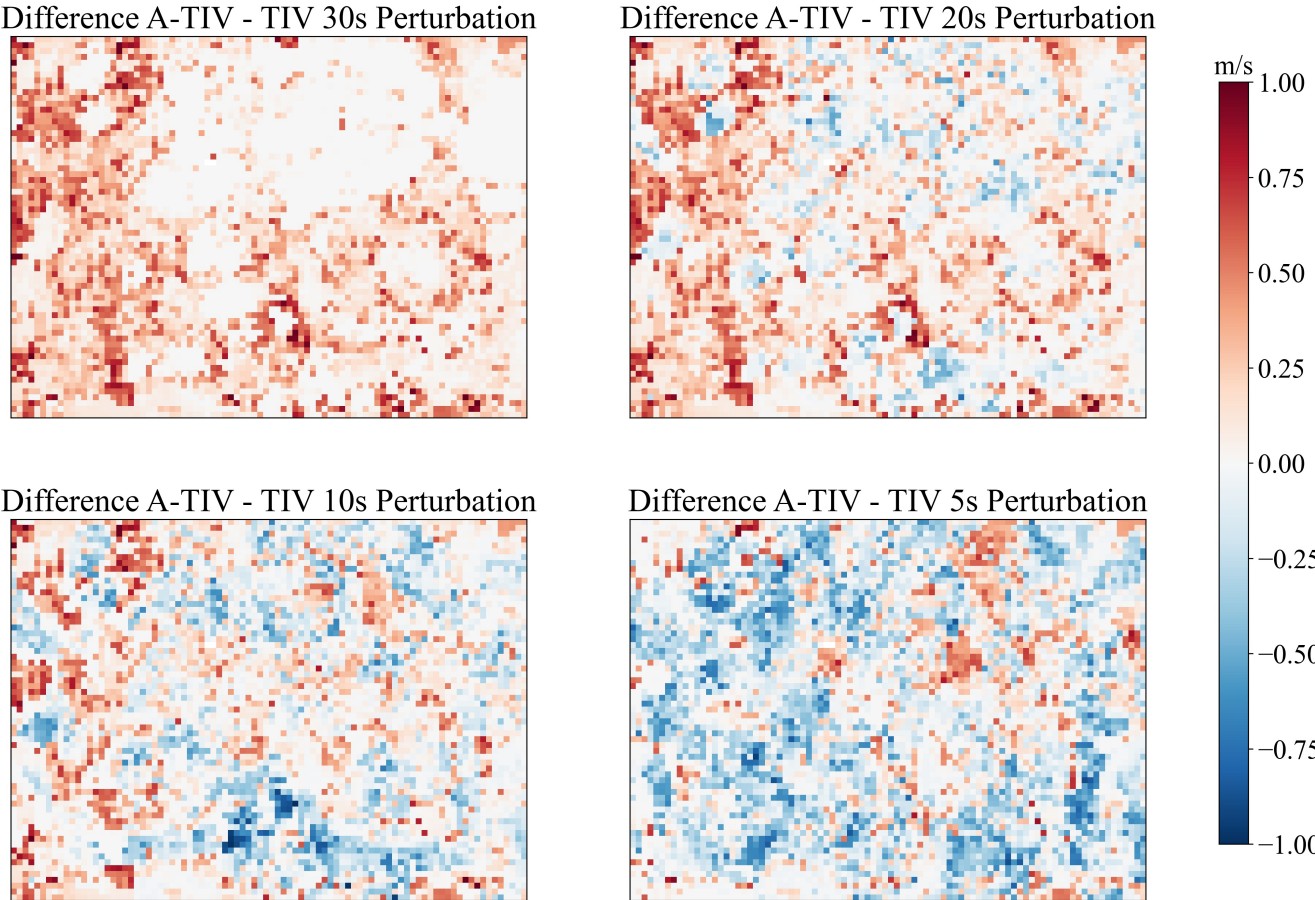

**Figure 8. A-TIV wind speed difference to the TIV result of TURF-T1 for an example frame** - Red indicates that the A-TIV result exhibits a higher speed than the corresponding TIV result. A positive difference means that all the other TIVs add velocity information to these pixels according to their weights (respectively meaning no velocity information is gained from this TIV pixel). A negative value means that velocity information is added from this pixel. A value close to zero means that the TIV of this pixel is matching the A-TIVs value. All velocity information gains and losses are changes in respect to the weight which is defined by the perturbation filter time. A-TIV is therefore gaining information from different TIVs and creates an average measurement over various perturbation scales.

Table 3 shows the quantitative advantage of A-TIV over TIV. A-TIV provides less vacant grid cells compared to any TIV with a fixed perturbation filter size. Most of the velocities are added by the TIV with the highest perturbation filter size, whereas the other TIVs add less information to the final A-TIV product. The percentage given is the number of cells that add velocity information respective to the full amount of available cells. Therefore, each TIV product could reach up to 100 % of added information.

**Table 3.** Comparison of A-TIV with TIV - further qualitative information also available in appendix B

|  | A-TIV | TIV-30s | TIV-20s | TIV-10s | TIV-5s |
|---|---|---|---|---|---|
| Vacant Grid cells | 0.4 % | 24.2 % | 19.9 % | 11.2 % | 4.6 % |
| A-TIV cells gain velocity information from this TIV in % | - | 86.3 % | 67.1 % | 51.4 % | 36.4% |
| Average A-TIV/TIV speed (averaged for TURF-T1 over entire area) | 0.35 m/s | 0.43 m/s | 0.33 m/s | 0.25 m/s | 0.22 m/s |
| RMSE of A-TIV direction compared TIV direction (averaged for TURF-T1 over entire area) | - | 14.2 ° | 12.7 ° | 11.2 ° | 14.1 ° |
| Weight of the TIV |  | 6 fold | 4 fold | 2 fold | 1 fold |

### 3.2.3 Evaluation of A-TIV with wind velocity measurements

The A-TIV velocity fields for TURF-T1 were compared in two steps with the *in-situ* available sensors the sonic anemometer and the lag-correlated thermocouple wind speeds. The first comparison covered the magnitude and timing of the perturbation scales of temperature and wind speed (Figure 9). It showed that the A-TIV measurement matches the magnitudes of wind speed
perturbation very well, when averaged over an area. There is also a clear timing difference of the perturbation peak occurrences between the A-TIV and the sonic anemometer data which is expected due to the differences in measurement height and type.

     The second comparison also involved the turf and grass surface within TURF-T2 experiment. This comparison revealed the different impacts of the surface types to the A-TIV measurement. In general, the TC wind speed distribution is shifted to higher wind speeds due to the limitations of the method allowing only wind speeds greater than 0.25 m/s. Compared to the
spatial TC wind speeds, A-TIV is measuring a similar wind speed distribution on the artificial surface type of TURF-T1 which is confirming the results presented by Inagaki et al. (2013) (Figure 10). With TURF-T2 a new mixed surface type was added and A-TIV resolved generally higher wind speeds than the TURF-T1 experiment which aligns the mean wind speed measured by the sonic anemometers (Table 1). The mean of the A-TIV speed distributions of TURF-T2 is 2.2 times higher than the TURF-T1 mean which is a similar ratio as the average of the sonic anemometer mean wind speeds from both experiments
(Table 4).

     Figure 11 shows the relationship between A-TIV speeds compared to sonic anemometer wind speeds and TC wind speeds. The results are comparable with previously published relationships by Inagaki et al. (2013) over artificial surface types. Notable is that in the TURF-T2 experiment there is no significant relationship between the aereal average A-TIV speed and the wind speeds from the sonic anemometers.

**Table 4.** Sonic Anemometer mean wind speed compared to the A-TIV mean speed

| Experiment | Mean wind speed | Mean A-TIV speed | Ratio |
|---|---|---|---|
| TURF-T1 | 2.60 m/s | 0.43 m/s | 6 : 1 |
| TURF-T2 turf | 5.86 m/s | 1.03 m/s | 5.7 : 1 |
| TURF-T2 grass | 5.71 m/s | 0.91 m/s | 6.2 : 1 |
| Wheat stubble | 8.41 m/s | 0.55 m/s | 15.3 : 1 |

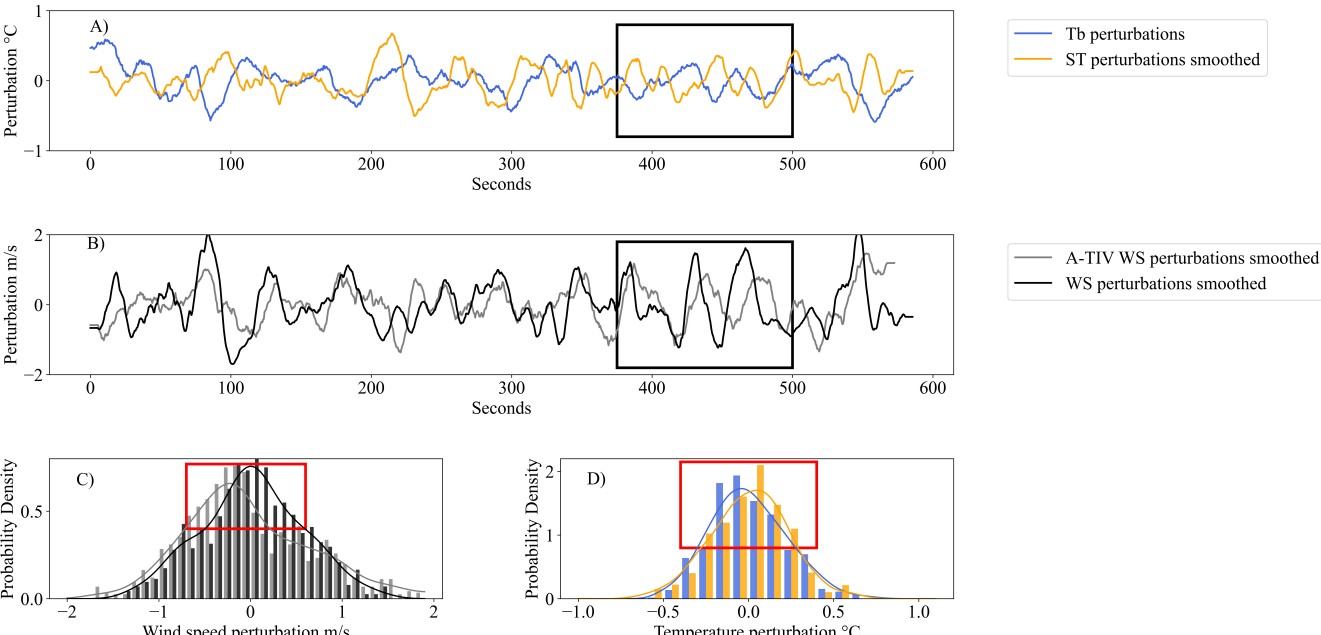

**Figure 9. TURF-T1 wind speed and temperature perturbations** - TURF-T1 Comparison of spatial average (15 m x 15 m) brightness temperature (Tb) perturbations ( A) blue) with Sonic Temperature (ST) perturbations ( A) orange) and the areal A-TIV wind speed perturbations ( B) gray) with the measured wind speed perturbations from the eddy covariance/sonic anemometers ( B) EC - black). A) and B) shows that the timing of the measured temperature perturbations - see black square. The red rectangles in B) and C) show the differences in distribution of perturbations. The area ∼ 15 m x 15 m for the calculation of the spatial average perturbations was picked slightly upwind of the sonic anemometer.

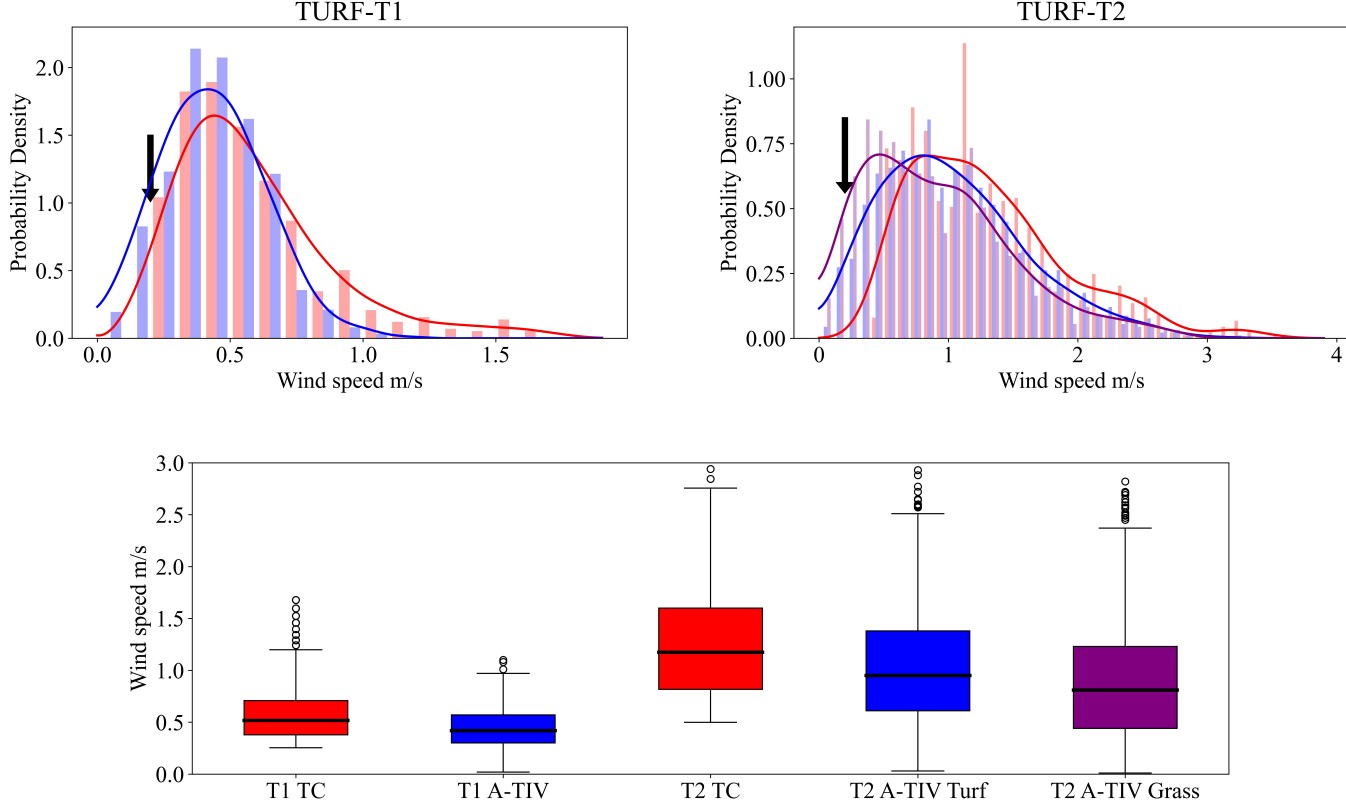

**Figure 10. TURF-T1 and TURF-T2 accuracy assessment with TC array** - TURF-T1 and T2 wind speed histogram comparison. Blue histograms are A-TIV histograms of an area of $\sim$ 15 m x 15 m, the purple histogram is showing the same area over grass, red histograms are thermocouple wind speeds. The black arrows mark the cut-off point of the minimal resolvable wind speed of the lag-correlation method from the thermocouples. The boxplot on the bottom shows the differences in the median value of the displayed histograms. Across all surfaces the difference of the median is within $\pm$ 0.4 m/s.

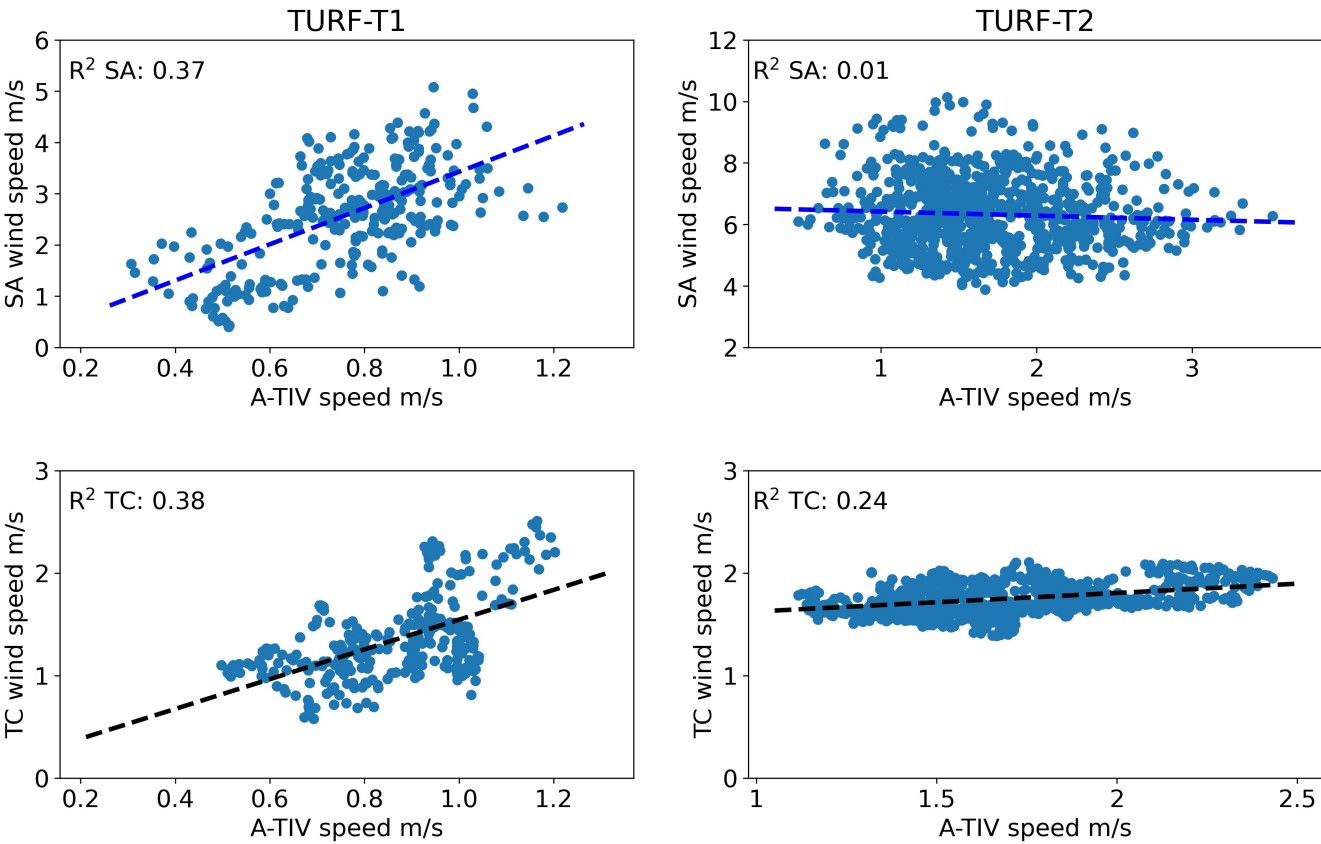

**Figure 11. A-TIV speed relationship with other measurement methods** - TURF-T1 and TURF-T2 relationship of wind speeds measured by the sonic anemometer and the lag-correlation of the TC array with the A-TIV speed. It is evident that TURF-T1 shows a higher $R^2$ values than TURF-T2. A weak relationship between the sonic anemometer and the A-TIV speed in TURF-T2 is evident.

### 3.2.4 Evaluation of A-TIV for higher roughness elements (wheat stubble)

The wheat stubble A-TIV was carried out to test the limitations of the A-TIV algorithm when a higher canopy ($\sim 20$ cm) which is not affected by wind induced movements, is present. The effect of different perturbation temporal running filter size on the averaged vector field show that the assimilation is increasing the number of higher wind speeds due to the shortest perturbation window (Figure 12). However, through the weighted averaging in the A-TIV calculations the shortest perturbation window does not have a large impact on the combined final wind speeds (Figure 12). Due to a fault in the high-frequency wind velocity measurements in the wheat stubble experiment the A-TIV based advection velocity measurement can not be compared to high frequency *in-situ* measurements and is therefore only of qualitative nature. Nevertheless, when comparing the combined histograms from the A-TIV measurement of all experiments we hypothesise that the canopy height is causing a decrease in measured A-TIV based advection speed due to the influences of drag introduced by the canopy (Figure 13). This is supported by the average wind speed ratios of the three experiments which would also lead to a higher expected A-TIV velocity estimation (Table 4).

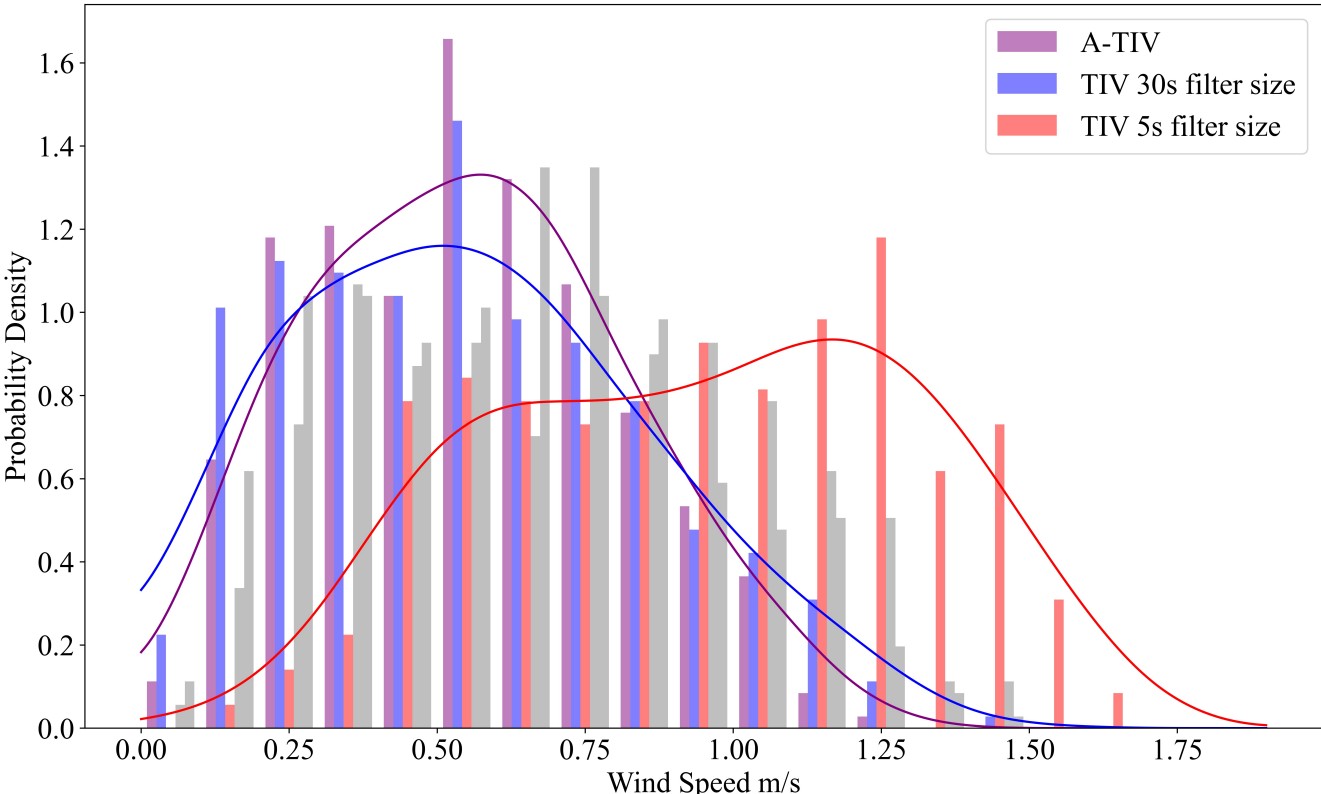

**Figure 12. Impact of the lowest and highest perturbation running filter size on the wheat stubble A-TIV result** - The blue histogram is based on the 30s perturbation window, the red histogram on the 5s perturbation window. The combined final wind speed histogram is shown in purple. The perturbation temporal running filter sizes 10s and 20s are shown in grey.

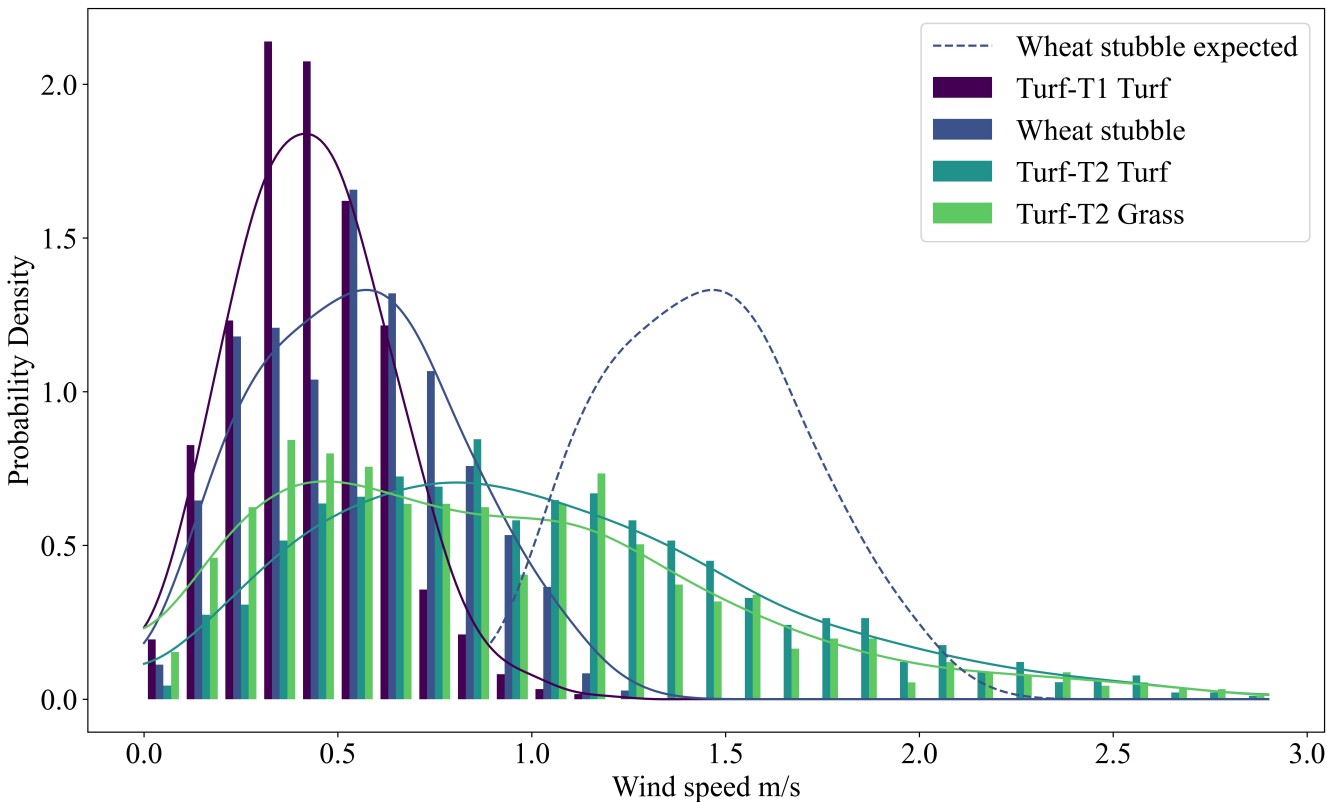

**Figure 13. A-TIV comparison of all experiments** - Probability Density functions for the $\sim$ 15 m x 15 m average aerial wind speed from all experiments. It is evident that the Wheat stubble experiment is showing the lowest average wind speed histogram distribution whereas the mean wind speed was the highest during the experiment. The dashed line shows the expected probability density based on the ratio 6:1 calculated in Table 3. The decrease demonstrates the effect of the canopy.

## 4   Discussion

A-TIV is a first effort to calculate a spatial atmospheric velocity measurement over natural surface types and make the algorithm available to a wider community in atmospheric science and near-target infrared remote sensing. Garai and Kleissl (2011)
pointed out that movements of leaves or the movement of the canopy itself within the field of view of the sensor itself can influence the brightness temperature measurement. This can impact near-target infrared measurements as well depending on the canopy height and the wind velocity (Garai and Kleissl, 2011; Finnigan, 2010). Due to the harvested wheat pasture such effects are not expected for the wheat stubble experiment. However, the higher canopy of the wheat stubble has an additional insulation effect by absorbing thermal radiation. This radiation is emitting over longer periods of time whereas short turf or
grass is emitting the longwave thermal radiation faster. The effect of slower emittance over wheat stubble interferes with the brightness temperature perturbations leading the A-TIV algorithm to estimate a low velocity distribution while higher mean

wind velocity was measured (Figure 13 and Table 1). Furthermore, the wheat stubble alters the wind profile above it more significantly compared to the smooth roughness elements of TURF-T1 and TURF-T2 which also affects the exchange of heat between the surface and the atmosphere.

Nonetheless, there are additional limitations involved when collecting brightness temperature from a UAV. To avoid noise and temperature spikes from external solar heating of the camera body the Optris camera provides a resetting mechanism every twenty seconds for one second. During this period, the camera is not acquiring images which then must be removed from the time sequence and possibly cause the timing delay of the calculated A-TIV velocity perturbations (Figure 9). Furthermore, due to flight height restrictions of the UAV a wide-angle lens was used to cover a larger area. This implies a distortion of the camera

lens which has not been removed because the stabilization and the A-TIV algorithm involve cropping the edges of the images. An assessment of the pixel distortion showed that for the TURF-T1 experiment the maximal deviation from the calculated pixel size was 0.05 m. This error is within the error range of the TC array calculated wind speeds which can be affected by the possible misplacement of the sensors on the ground. For TURF-T2 the distortion was further minimized by a lower flight height of the UAV and hence a smaller field of view was covered (Table 1).

The final 2D velocity vector from A-TIV represents the interaction of atmospheric coherent structures with the surface. Structures can have a higher or lower velocity than suggested by the mean wind if the advection is not purely horizontal. Hence, A-TIV is not necessarily a direct measurement of the mean wind of the coherent structure causing the A-TIV velocity estimation. Furthermore, the estimations depend on the scale of the interaction in time and space. Larger structures than the defined correlation window size as well as small structures smaller than half of the correlation window size might not be

resolved by the A-TIV algorithm.

An advantage of the A-TIV algorithm is that is provides a spatial measurement of wind velocities at the resolution of the camera sensor array. This is an advantage to traditional point-based measured velocities which are limited to a spatial footprint dictated by the height of the sensor on a tower and the mean wind speed. The spatial near-surface velocity measurements through A-TIV can give valuable insight into the instantaneous turbulent eddy sizes and future work can focus on evaluating

the resolved turbulence spectra and compared to spatial eddy covariance measurement campaigns. This is specifically advantageous when combining the two measurement approaches to further analyse the point measurements which are depending on Taylor's Hypothesis and are potentially prone to underestimating the turbulent spectrum in the inertial subrange (Taylor, 1938; Engelmann and Bernhofer, 2016; Cheng et al., 2017)

Furthermore, the A-TIV measurement may not reflect the same atmospheric turbulence compared to the sonic anemometer

mounted at 1.5 m height. Previously Inagaki et al. (2013) applied a correction factor to the TIV measurements to compare them to sonic anemometers. This correction factor was not applied in this study because the measurement footprint of the sonic anemometer was larger and the $R^2$ values specifically in TURF-T1 show similar correlation as determined by Inagaki et al. (2013).

The thermal interactions of structures with shorter time scales create a reduced amplitude in thermal perturbations measured

by a sonic anemometer or measured by thermal imagery. Without a time-frequencey decomposition on the signal, the sonic anemometer registers these as one time series integrating these scales and represented by a measured temperature or wind

velocity. The TIV does not directly reflect this frequency composition and may reflect only certain frequencies due to the decomposition done in the calculation of the perturbation and the Hilbert-Huang Transform. Higher frequencies and sensor noise are neglected in this way. To mitigate this indirect focus and limitation to certain frequencies we introduced the A-TIV composition of multiple perturbation windows which then allows to reflect the reality better which is a composition of superimposed multiple frequencies. In our case we decided to use 30, 20, 10 and 5s because our thermal sensors noise was only starting to interfere with the 5 s perturbation window. Therefore, we decided to use a weighted averaging system because it is expected that the noise level rises with the decrease of the temporal perturbation filter size. This is the case for any sensor used and is not limited to the thermal camera.

For any lower quality camera systems that includes more noise in the first detected frequency by the HHT, the second highest frequency may be picked. This may be also successfully used, however the A-TIV output may display more vacant grid cells.

A-TIV and TIV have only been tested in dry, warm environmental (approximately 20°C) conditions with low latent heat flux present. Moisture, high roughness elements and shading directly lower the fluctuations in the brightness temperature measurement caused by atmospheric-surface interactions. Further assessment is needed to test the range of environmental condition A-TIV can be applied to such as below 0°C conditions. It is expected that this is mainly dependent on the amount of latent heat flux, the surface type, the canopy height, and the accuracy of the camera.

When comparing A-TIV measurements to Eddy Covariance measurements in TURF-T1 (Figure 9) there is an indication for a coupling between the surface and the air temperature. But neither the wind speed perturbations nor the temperature perturbations show similar distributions. However, both measurements show similar magnitudes of perturbations. This emphasizes that the A-TIV captures cooling and heating patterns when the atmosphere is interacting with the surface. However, the histograms show that the distributions are not comparable, which is expected comparing a point measurement to a spatial approach. This means that the A-TIV is reflecting a spatial measurement whereas the other measurement methods are based on single point measurements which depend on their mounting height respectively their footprint. The direct spatial measurement of A-TIV reflects the atmospheric situation directly adjacent to the surface and hence, when compared to point measurements further away from the surface, may not reflect the same conditions. The patch size of 15 m by 15 m to compare A-TIV to the Eddy Covariance measurements was chosen based on the size and the calculated footprint of the available upwind area (25 m x 20 m) of the sonic anemometers in the TURF experiments (see appendix on footprint calculation). To avoid corner effects i.e. from obstacles and to reflect the core of the calculated footprint we decreased the size of the averaging area to 15 m by 15 m.

The assumption from Stull (1988) that meteorologists rather observe atmospheric conditions over longer periods of time (> several hours), than creating short observations over a large region of interest does not reflect the strategy of A-TIV applications. According to Stull (1988) the long term point measurements can be translated to their corresponding spatial measurements as a function of time. A-TIV is a new approach in the sense that the measurement type is directly spatial and hence short term observations can immediately reflect the spatial component of turbulence. Moreover, the new type of data that is retrieved needs new spatiotemporal statistics and new anaylsis methods such as A-TIV for new insights into spatial turbulence.

Essentially the underestimation of A-TIV velocity measurements is a result of the difference in the measurement process between the TC lag-correlation and the A-TIV. A-TIV will resolve only wind velocities when the coherent structures exhibit

a temperature difference to the surface temperature. Therefore, A-TIV is fully based on the difference of spatial change in brightness temperature, whereas the lag-correlated TC wind speeds are estimating motion based on air temperature changes measured in points and are not bound to the thermal properties of the surface. This means that the A-TIV algorithm will resolve surface atmospheric interactions and motions of a certain scale very well, however very small-scale processes within the size of half of the correlation window size will not be resolved by the algorithm.

Specifically small spatial interactions with low velocities may not be reflected in the TIV estimations with a higher temporal running filter size (in this manuscript 30s and 20s). Therefore, the A-TIV includes velocities from lower temporal running filter sizes (10s and 5s in this manuscript) and ensures that less vacant grid cells are present compared to any TIV. The display of very small velocities ($< 0.5$ m/s ) is also not ideal due to the a high range of extracted velocities from the multiple TIVs neglecting the display of small velocities (Figure 7).

It is evident that A-TIV speeds are related to the wind velocity measured by other measurement techniques (Figure 9). However, when the wind speed increases the relationship between the sonic anemometers and the A-TIV speeds is not as strong. This is likely due to the elongation of the thermal structures which has been described before (Garai et al., 2013). Streaky surface structures are measured differently with the thermal camera compared to the sonic anemometer which was mounted at 1.5 m height. More detailed investigations are needed to quantify the measurement differences and possibly the adjustments to the A-TIV algorithm to better resolve the surface layer turbulence.

## 5  Conclusions

The results from this research present an enhancement of the TIV algorithm the Adaptive Thermal Image Velocimetry which enables it to derive spatial wind velocity measurements from thermal images at moderately high frequency (2 Hz) over artificial and non-artificial surfaces.

The key findings of this study are:

1. High frequency ($>1$ Hz) brightness temperature measurements over dry thermally responsive surfaces reflect similar atmospheric influences as near surface (approximately 1.5 cm) temperature measurements (Figure 5).

2. Brightness temperature measured with a UAV, when software image stabilization applied, can be used to retrieve instantaneous spatial wind fields using A-TIV over artificial and non-artificial surface types (Figure 10).

3. The TURF-T experiments showed that A-TIV can correctly resolve air temperature perturbation and wind speed perturbations and retrieve spatial velocity fields very close to the ground (approximately 1.5 cm) when compared to *in-situ* lag-correlated thermocouple measurements (Figure 9 and Figure 10 ).

4. The wheat stubble experiment showed the impact of the canopy height on the A-TIV wind speed distribution. Further investigation is needed to evaluate the impact of the canopy height on the algorithm settings (Figure 13).

The key limitations for a successful retrieval of higher frequency A-TIV velocities are the camera's noise levels and the spatial field of view of the camera. This means that small infrared cameras carried from UAVs are delivering high acquisition rates (Optris Pi 450: 80 Hz), however due to the camera noise present at this frequency, need average sub-sampling to lower frequencies to enable analysis on the brightness temperature imagery. This is in direct connection with the camera's distance to the target surface where the acquired temperature differences become less due to the physical property of infrared light carrying less energy than visible light. This creates a feedback with the electrical noise of the camera. Therefore, depending on the acquisition situation we suggest a careful assessment of all environmental parameters to retrieve optimal results. In similar conditions as presented in this study, a sub-sampling averaging to $<3$ Hz is recommended.

The potential of thermal cameras in remote sensing of micro-meteorology is large and the A-TIV algorithm can be seen as a new opportunity for advanced analysis methods of spatial velocity field measurements. Further investigation will be needed to optimize the algorithm's performance and usability especially over new surface types to make it available for a larger community of remote sensing specialists and atmospheric scientists.

*Code availability.* The open-source code to use TIV and A-TIV is available on Github (Schumacher, 2021)

*Sample availability.* The Github repository also contains data samples from the presented experiments (Schumacher, 2021)

**Appendix A: Appendix A - Footprint Calculation**

The footprints used for comparison of the sonic anemometer measurements with the A-TIV algorithm were calculated using the Urban Multi-scale Environmental Predictor (UMEP) (Lindberg et al., 2018) with input data from Land Information New Zealand and the measured variables. The footprint model used in the calculation was set to Kormann and Meixner (2001). For TURF-T1 and TURF-T2 the averaging area (15 m by 15 m) was picked within the field of view of the infrared camera and covered $\sim 60$ % of the cumulative source area (see Figure A1 for an example from TURF-T1).

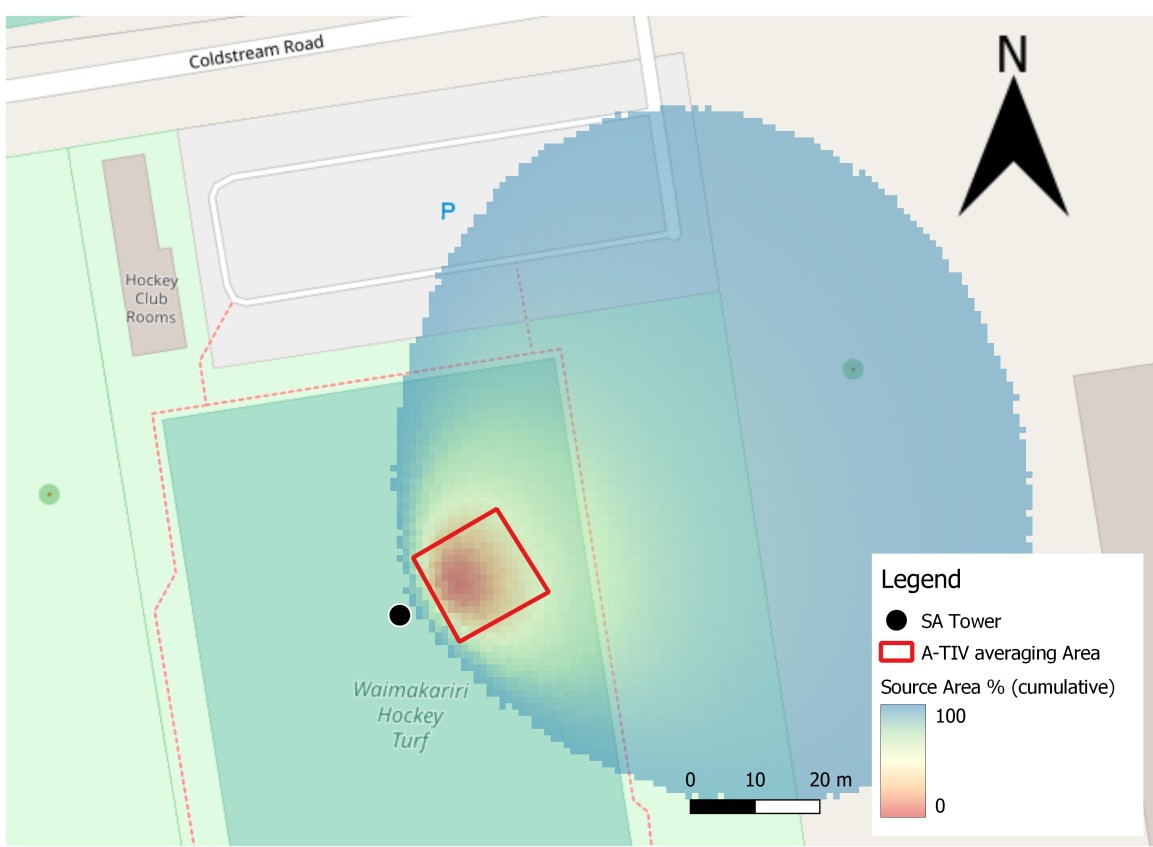

**Figure A1. Footprint calculated from the TURF-T1 experiment** - Background Image: © OpenStreetMap contributors 2022. Distributed under CC BY 4.0 licence

## Appendix B:  Appendix B - A-TIV - TIV windspeed and wind direction comparison

The qualitative comparison of A-TIV wind speed and wind direction with TIV wind speed and wind direction is shown in Figure B1 and Figure B2. While the TIV wind speed depends mainly on the perturbation filter sizes the A-TIV reflects the weighted mean of all caluclated TIVs and removes outliers that may depend on camera noise or short temporal thermal disturbances within the measured thermal perturbation. On the other hand the measured wind direction signal of A-TIV compared to TIV is very similar. This shows that both methods are reflecting directions of moving thermal patterns correctly.

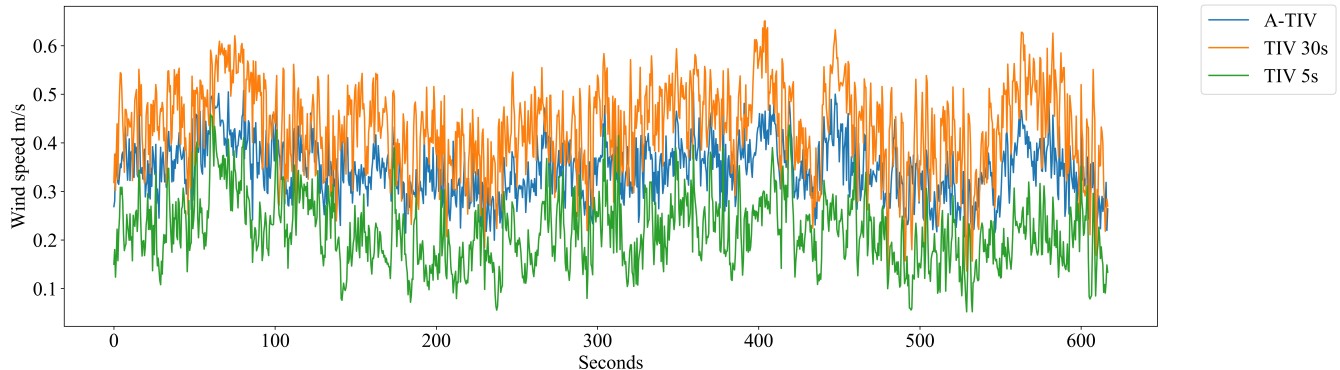

**Figure B1. Qualitative comparison of A-TIV wind speeds to TIV wind speeds of TURF-T1** The speeds are averaged over the entire area of the turf field. TIV speeds from the perturbation filters 10 s and 20 s would be located between the 5s and the 30 s wind speeds and are not shown in this plot.

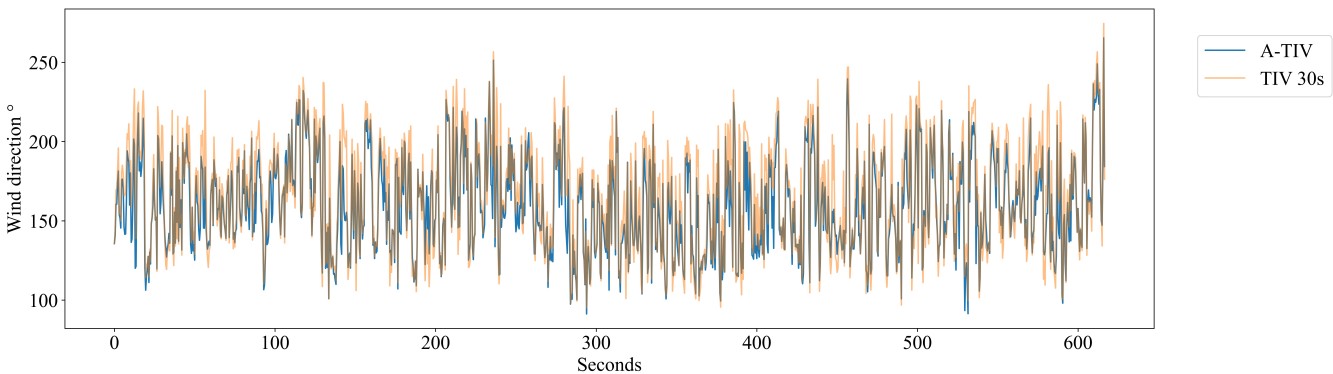

**Figure B2. Qualitative comparison of A-TIV wind directions to TIV wind speeds of TURF-T1.** The directions are averaged over the entire area of the turf field. TIV directions and A-TIV directions are very similar, hence this plot shows only the 30 s TIV directions and the A-TIV directions. For a quantitative comparison please refer to table 3

*Author contributions.* Matthias Zeeman was involved in the conceptual framework of the experiments and the infrared data acquisition and analysis. Benjamin Adams supervised the software development which was carried out by Benjamin Schumacher. Jiawei Zhang was involved in the conceptualization of the stabilization. Peyman Zawar-Reza and Marwan Katurji are responsible for the funding acquisition. Benjamin Schumacher and Marwan Katurji designed and conducted the experiments and analysed the data. Benjamin Schumacher prepared the manuscript with contributions from all co-authors.

*Competing interests.* There are no competing interests present.

*Acknowledgements.* This work was carried out under the framework of the research program "The invisible realm of atmospheric coherent turbulent structures: Resolving their dynamics and interaction with the Earth's surface" which was funded by the Royal Society of New Zealand with contract number RDF-UOC1701. We want to express special thanks to Justin Harrison, Paul Bealing and Nicholas Key who helped with the TURF experiments and the UAV operations. A special thanks to the Scion field crew for supporting the wheat stubble experiment: David Glogoski, Max Novoselov, Richard Parker, Brooke O'Connor, Emma Percy, Ilze Pretorius, Veronica Clifford, Jessica Kerr, Kate Melnik and Grant Pearce.

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
