# Peer review of "Adaptive Thermal Image Velocimetry of spatial wind movement on landscapes using near target infrared cameras"

_Atmospheric Measurement Techniques, 2021_

## Author Response (AR1)

Reviewer 1:

In the article, authors have described a technique to estimate the wind velocity from thermal images. The estimated wind velocity is comparable with the wind speed measurement qualitatively. But quantitatively, e.g. $R^2$ analysis, p-value) the agreement between them is poor. Also many details of the analysis are missing. Thus, I won't recommend the paper for publication in current form. Following are the itemized suggestions:

Answer:

Thank you for your detailed suggestions and comments. We have updated the manuscript according to your concerns.

Reviewer 1:

1) How authors calibrate each pixel of the thermal image to physical distance, m? The pixel sizes vary based on the camera viewing angle. Pixels closer to the camera have smaller physical sizes compared to the further away. How authors have corrected this camera viewing distortion of the thermal image? What are the image sizes in physical distance?

Answer:

As stated in Line 321 referred to the revised track change version here and after, distortion showed the maximal deviation the calculated pixel size was 0.05 m. This means that a pixel closer to the camera in the center of the image is maximally 0.05 m smaller in x- and y-direction than a pixel on the edge of the area of interest. The size of the hokey field is 91.4 x 55 m. The deviation was calculated using the number of pixels between the midfield line and the 22.85 m line and the latter one and the goal line for the x-direction. The same was calculated for the y-direction using the pixels located at the goal lines and at the center line. The pixels on the very edge of the field were excluded from the calculation as stated in line 345. We have further clarified this paragraph (line 336 - 340) and added the size of the areas of interest to the corresponding experiment overviews (Table 1).

Reviewer 1:

2) How did authors perform the spatial average of the 15m x 15m patch? From figure 5, the velocities are estimated when temperature gradient is present. Does the patch size ensure no missing wind velocity estimate for the pixels considered? Why specific size of patch is chosen?

Answer:

The average was performed with a spatial mean function over the 15m x 15m patch. The A-TIV method ensures that less vacant grid cells are calculated compared to any TIV performed (see Figure 5b and improvement in vector spatial density relative to the missing values in figure 5a). The vacant grid cells of the TIVs resulting from higher perturbation filters are adapted to the predominant perturbation time scale by the weighted average resulting in the A-TIV output velocity grid. Therefore, the vacant grid cells of higher perturbation TIVs (30s and 20s) are not resulting in 0 or NA values for the spatial averaging. Small velocities (< 0.5 m/s) may not be displayed with the vectors in the figures. The patch size was chosen based on the available upwind area of the sonic anemometer within the field of view of the camera and the size and the calculated footprint (see answer to question 3 for the footprint calculation). To avoid corner effects i.e. from obstacles and to reflect the core of the calculated footprint, we decreased the size of the averaging area to 15 m x 15 m. We have added a clarification to the manuscript (line 367-370 and line 393-401) alongside with the new calculations of the flux footprint in response to reviewer question 3 below).

Reviewer 1:

3) For comaprison with the sonic anemometer measurement, how the location of 15m x 15m patch is chosen? Does it lie inside the footprint of sonic anemometer? If not, why authors have not considered some flux footprint model, as descibed in Garai et al. 2013, Boundary Layer Meteorology?

Answer:

We have added new material (Appendix) for the estimation of the footprint of the sonic anemometer using the UMEP plugin for QGIS which allowed us to pick a suitable area for the averaging (Lindberg 2018). We have now prepared a more detailed view in the appendix. We have also referenced the Appendix in the text where needed.

Reviewer 1:

4) What are the weights for averaging wind velocities from different perturbation? How the weights are chosen?

Answer:

The TIVs within the A-TIV process resolve various perturbation scales (30s – 5s in the present manuscript). With decreasing perturbation time scale the amplitude of the perturbation is decreasing and the noise level of the camera is becoming more present in the perturbation signal. Therefore, we have decided that the resulting TIVs are weighted according to the perturbation scale its calculation is based on. In our case this is 30s – 20s – 10s and 5s in this manuscript meaning weights of sixfold, fourfold, twofold and onefold (based on perturbation filter divided by 5). We have rephrased the paragraph explaining this procedure in line 129 – 134.

Reviewer 1:

5) Figure 7, how the difference is calculated? Not all the pixels result velocity estimation.

Answer:

The difference velocity fields were based on pixel to pixel subtraction. A-TIV creates an velocity estimation for each pixel. This may be that the velocity is 0 and hence no difference is estimated (white pixels). The purpose of this figure is to show the added information associated with the ATIV approach relative to the perturbation average window. Through the weighted averaging (see question 2), more pixels are resulting in a velocity estimation compared to any single TIV. Additionally, the differences between A-TIV and a single TIV result shows the need for the A-TIV.

Reviewer 1:

6) Lower value of p-estimation means that the assumed null hypothesis does not hold. For present study, what is the null hypothesis, is it TIV corresponds to wind velocity? If so, then the reported small p-value means there is no correlation.

Answer:

Your statement is correct. This analysis used the null hypothesis (H0) that the means of the two distributions do not match. Hence the low p-value. We have adjusted H0 and report now a higher p-value as per common practice (line 273 - 275).

Reviewer 1:

7) Figure 9a. Too many wiggly lines make it difficult to read. Authors should consider separate out the temperature and velocity curves in two separate figures.

Answer:

We have adjusted the figure and separated the lines into two plots.

Reviewer 1:

8) Figure 9 shows that the TIV and sonic anemometer have some temporal lag. It also looks like the lag and wind direction are not constant for the time period considered. How the authors account for these effects when calculating $R^2$, p-value and histograms for quantitative comparison? Also how these variable effects the comparison.

Answer:

Figure 9 may show a temporal lag due to multiple reasons:

1) As mentioned in line 342 - 343 this lag may be caused by the resetting mechanism of the thermal camera.

2) Essentially the sonic anemometer and the A-TIV do not necessarily reflect the same atmospheric turbulence. While the sonic anemometer mounted at 1.5 m height reflects a larger footprint the A-TIV only measures in-situ temperatures and spatial perturbation changes between images with the likelihood of measuring structures very close to the surface (within centimetres above the surface).

The p-value expresses only statistical (distribution) relationships between the measurements and not direct correlations. It is shown in the manuscript that the thermal imagery and A-TIV reflects the turbulent flow ~1.5 cm above ground whereas the sonic anemometer reflects a point measurement at 1.5 m above ground with a spatial footprint. Hence a direct correlation value such as the $R^2$ is lower because the A-TIV measurement cannot directly explain all atmospheric turbulence measurements of another sensor of a different type. We have added a dedicated paragraph in the discussion section explaining this (line 367-370).

Reviewer 1:

9) Figure 10, What are the markers in the figure?

Answer:

The Figure caption states that the black arrows mark the "minimal resolvable wind speed of the lag-correlation method from the thermocouples"

Reviewer 1:

10) Page 17, line 261: What do you mean by positive wind speed? Wind speed is always positive.

Answer:

We have rephrased the corresponding sentence (line 307).

Reviewer 1:

11) For Turf-T2 why authors have not considered to have two thermocouple arrays on the two surfaces, instead of putting one thermocouple array in the mixed surface. As the surface properties are changing, a new boundary layer will start to develop. How authors account for that in the analysis.

Answer:

Due to limited amount of Thermocouples it was not possible to create two separate arrays. The Analysis shows that the lag correlations performed between temperature perturbations measured with "physical" thermocouples and with "virtual" thermocouples with the same surface cover result in very similar estimations. However, when the virtual array was moved in

either surface cover the estimations no similarity was determined. Therefore, it was decided to separate the A-TIV calculations for both surfaces. An explanation has been added to the methods section (line 255-256).

Reviewer 1:

12) Garai and Kleissl 2013, Journal of Turbulence, reported that the different temporal filtering result thermal structures corresponding different scale. How authors account for that when comparing averaged TIV from different temporal perturbation with sonic anemometer? The small flow structure giving the TIV 5s perturbation may not be registerd at the sonic anemometer.

Answer:

The thermal interactions of structures with shorter time scales create a reduced amplitude in thermal perturbations measured by a sonic anemometer or measured by thermal imagery. Without a time-frequencey decomposition on the signal, the sonic anemometer registers these as one time series integrating these scales and represented by a measured temperature or wind velocity. The TIV does not directly reflect this frequency composition and may reflect only certain frequencies due to the decomposition done in the calculation of the perturbation and the Hilbert-Huang Transform. Higher frequencies and sensor noise are neglected in this way. To mitigate this indirect focus and limitation to certain frequencies we introduced the A-TIV composition of multiple perturbation windows which then allows to reflect the compositions of multiple frequencies. In our case we decided to use 30, 20, 10 and 5s because our thermal sensors noise was only starting to interfere with the 5 s perturbation window. Therefore, we decided to use a weighted averaging system because it is expected that the noise level rises with the decrease of the temporal perturbation filter size. This is the case for any sensor used and is not limited to the thermal camera. This paragraph has also been added to the manuscript alongside with further comments on the running temporal filter size (line 372 – 384)

Reviewer 1:

13) English in the article is poor.

Answer:

Thank you for all your comments and suggestions. We have corrected multiple grammatical, wording and spelling mistakes.

References:

Lindberg F, Grimmond CSB, Gabey A, Huang B, Kent CW, Sun T, Theeuwes N, Järvi L, Ward H, Capel-Timms I, Chang YY, Jonsson P, Krave N, Liu D, Meyer D, Olofson F, Tan JG, Wästberg D, Xue L, Zhang Z (2018) Urban Multi-scale Environmental Predictor (UMEP) - An integrated tool for city-based climate

services. Environmental Modelling and Software.99, 70-87
https://doi.org/10.1016/j.envsoft.2017.09.020

Review: Adaptive Thermal Image Velocimetry of spatial wind movement on landscapes using near target infrared cameras

The authors present a modification of the thermal image velocimetry (TIV) method called adaptive TIV (A-TIV). They use fluctuations in surface brightness temperature derived from time series of UAS-based thermal imagery for estimating two- dimensional near surface wind velocities.

With my background (I use UAS-based thermal imagery for assessing turbulent energy fluxes), parts of the manuscript remain unclear. The description of the method is not detailed enough in my opinion and the structure of the methods part is also a bit unclear to me (some sections would probably better fit into the results part?). The results part lacks important information, e.g. it does not provide any comparison of the presented A-TIV algorithm with the existing TIV algorithm, which would be essential for assessing the benefit of the new method.

Since I am not a native myself, I do not comment on language in general. But long sequences of nouns (e.g. multiple surface brightness temperature perturbation filter sizes) make the text hard to read, which might be avoided by rearranging sentences.

In my opinion this manuscript needs a thorough revision before publication.

I list some more specific comments below:

Answer:

> Thank you for your detailed suggestions and comments. We have updated the manuscript according to your concerns. We agree with the reviewer that more details of the method need to be provided. To address this issue, we have added an Appendix providing detailed information about the areal footprint of the sonic anemometer that, we now realized, are essential to include based on the reviewer's feedback. We also added a new table to compare TIV and A-TIV as well as new clarifications to questions raised about the methods. The discussion section was also expanded by several paragraphs to meet the reviewer's suggestions and discuss the newly added results.

Reviewer 2:

> P4: I would rephrase the objectiveimagerys since in the current form it is clear that the objectives were defined after conducting the experimimageryents as they already provide information about the outcomes.

Answer:

> We have rephrased the objectives and provided new and concise information about the outcomes (line 95).

Reviewer 2:

P5, line 118: I have no experience with the HHT, but what would happen if you chose the second highest frequency? The highest frequency is obviously the noisiest and might include camera noise effects?

Answer:

The highest frequency includes camera noise effects as well as usable signal. From experience with other (lower quality) IR cameras we have seen that the highest frequency contains camera noise and this approach may help (we have added this possible solution in the discussion section on line 384 - 387). However, with the equipment we used in the TURF experiments the HHT frequency decomposition helps in selecting the minimum time interval required for perturbation calculations that are used for the estimation of velocities from thermal imagery. This is the optimal approach because any other picked frequency would potentially miss higher frequencies. The second highest frequency may also be successfully used, however the A-TIV output may display more vacant grid cells.

Reviewer 2:

P 5, line 126: Add the weights here.

Answer:

We have added the weights according to your suggestion (line 128).

Reviewer 2:

P5, line 136: This sentence belongs to 2.1

Answer:

Thank you for the suggestion. We couldn't find line 136 on page 5. The lines around 136 also doesn't fit the reviewer's comments. We would kindly ask the reviewer to update us on the correct line number if this remains unresolved and requires further attention.

Reviewer 2:

P6: In part C the colors are not correct, it depicts twice the same 3x3 window. To me it is not clear why the correlation map has numbers in all pixels. A sentence on what the numbers in the correlation map mean would make it much easier to understand for people not familiar with the method.

Answer:

Thank you for pointing this out. We have corrected the figure with it's colours and added a sentence about the numbers in the correlation matrix. This correlation technique is also described in Schumacher et al. (2019), Inagaki et al. (2013) and Kaga et al. (1992). (line 137 - 139)

Reviewer 2:

P7, line 144: This sentence is totally unclear to me: *The weather station data is used to contextualize the A-TIV output in respect to the other experiments.*

Answer:

We have rephrased the sentence to clarify the use of the weather station data (line 150 - 152) It now reads: The weather station data was used to monitor the atmospheric conditions during the experiment and evaluate the A-TIV results in comnparison with the TURF-T1 experiment and TURF-T2 experiment.

Reviewer 2:

P 10, line 178: Did the scattered clouds have any effect on surface temperature perturbations?

Answer:

The scattered high clouds did not occur during the experimental period. Therefore, there were no effects on the surface temperature perturbations during the experiments. We have added this clarification to the manuscript (line 185)

Reviewer 2:

P10, table 1: what is the height of the grass? Grass can easily reach the same height as the wheat stubble. What is the ground resolution of the imagery? Why did flying altitude vary between the experiments?

Answer:

The grass was mowed and about 3-5 cm in height. We have added the information to table 1. The altitude varied to better resolve the small-scale turbulent structures in the grass and the turf area in TURF-T2 and the possible differences between them. Furthermore, the extent of the turf was smaller compared to TURF-T1 which limited the field of view of the camera and hence the flight altitude.

Reviewer 2:

P11, line 186: I do not see the cold spots. What is the emissivity of the high emissivity targets? Grass itself has a high emissivity. What is the approximate emissivity of turf? I am wondering why high emissivity targets are cold spots and not the low

emissivity targets? If I understand it correctly, this means that the air (the reflected part of the signal) is warmer than the surface, which drives a negative sensible heat flux? Could you explain this in more detail?

Answer:

Thank you very much for catching this important detail. The targets were polished aluminium plates (60 cm x 60 cm). Therefore, these targets had a lower emissivity compared to the surrounding turf and grass. We have corrected this mistake (line 195). In the figure caption of figure 4 the reference to the target is correct as "low emissivity targets".

In the previous text the high emissivity value for turf is mentioned as a requirement for TIV to work.

Reviewer 2:

P11, figure 4: Why is the peak in the standard deviation spatially shifted between a) and b)? It would make the interpretation of the images easier if also RGB images of the same scene were provided.

Answer:

Due to the shaking of the imagery the low emissivity target will shift from one video frame to the next. The Blender software tracking algorithm tracks this movement from frame to frame. Therefore, the spatial continuity is referenced to the first tracked frame not to the entire video sequence. This means, that the software will always try to match any new frame in the sequence spatially to the first frame of the video. A spatial continuity from unstable to stable imagery is based on the very first frame of the video not the entire video sequence. Hence the peak of standard deviation for the stabilized video will be at the first position of the detected low emissivity target. We have added the RGB imagery and the above explanation to the figure caption.

Reviewer 2:

P12, line 200: it is not clear to me how error vectors are assessed here?

Answer:

An error vector in this image is a vector which implies unrealistic effects such as localized large advection speeds. Most vectors in Figure 6 A) express 6-8 m/s in a very small area whereas the average wind speed during the day was 2.6 m/s. It is very unlikely that the A-TIV would cover such local wind speeds. We have added a clearer specification to the paragraph (line 281-282).

Reviewer 2:

P13 & 14, figure 6 & 7: I do not understand why these figures are part of the methods section? It would be more interesting to have something similar in the results section including a comparison of the different wind speed estimates with the reference data.

Answer:

We have moved the respective sections 2.5 and 2.6 to the results section becoming section 3.2.2 and 3.2.3. Furthermore, we have added an additional comparison table to section 3.2.2 which compares TIV to A-TIV in terms of vacant grid cells and added velocity information. This clarifies the advantage of A-TIV over TIV. The following sections show the comparison of A-TIV with the reference data.

Reviewer 2:

P15, line 223: Can you explain why you used a spatial shift of 9 m?

Answer:

The spatial shift of 9m was introduced to separate the virtual array completely from the physical array which has a 6m diagonal extent. Therefore, it was necessary to move the entire virtual array by 9m to the southwest/northeast to ensure no thermal signature is captured from the physical devices and wires.

Reviewer 2:

P15, line 233: how was the location of these 15x15 m windows selected?

Answer:

The reviewer 1 also asked the same question. Please see our answer to Reviewer 1 question 3. We have also added a section with the footprint calculation to a section in the amendments.

Answer to Reviewer 1:

"We have added new material (Appendix) for the estimation of the footprint of the sonic anemometer using the UMEP plugin for QGIS which allowed us to pick a suitable area for the averaging (Lindberg 2018). We have now prepared a more detailed view in the appendix. We have also referenced the Appendix in the text where needed."

Reviewer 2:

P16, line 249: I am confused concerning the p value. Is the null hypothesis that both data sets stem from the same distribution? Then this would be rejected for TURF-T1?

Answer:

Please see Reviewer 1 Question 6). We have adjusted the analysis to report a higher p-value as per common practice.

Answer to Reviewer 1:

"Your statement is correct. This analysis used the null hypothesis (H0) that the means of the two distributions do not match. Hence the low p-value. We have adjusted H0 and report now a higher p-value as per common practice (line 273 - 275)."

Reviewer 2:

P16, figure 8: In general, legends are missing in the figures. I would further encourage the authors to make their plots a bit more black/white friendly.

Answer:

Thank you for  bringing up the concern. We strongly agree with the reviewer that the plots should be easy to distinguish and colorblind-friendly. To be compliant with the journals' regulations we have checked each individual figure with the colorblindness simulator Coblis and the screen tool color oracle. All figures comply with the most frequent dispositions of colorblindness. The diverging colormaps in figure 5, 6 and 7 are compliant with the diverging colorblind colorbars from matplotlib python. See: https://github.com/matplotlib/matplotlib/issues/7081/

All figures except for Figure 8 expose legend items that help to understand the figures. Figure 8 does not necessarily need this legend item to express that there is no significant difference between the thermocouple and the brightness temperature measurement.

Reviewer 2:

P17 line 262: maybe I missed it but I think it was never stated before that the TC wind speed is limited to >= 0.25 m/s.

Answer:

We have added a corresponding sentence to the Methods section (line 239 - 240)

Reviewer 2:

P18 figure 9: this plot is a bit hard to read

Answer:

We have separated the line plot into two separate plots.

Reviewer 2:

P22, figure 13: which ratio is used for the dashed expected line?

Answer:

We have clarified the ratio in the figure caption.

Reviewer 2:

P22 line 291: this sentence is not clear to me

Answer:

We have restructured the sentence and split it into two separate sentences and added some additional information (line 336 - 340).

Reviewer 2:

P23 line 295: what would happen if the resetting mechanism is set to a longer interval? What would be the effect on the thermal patterns and on the absolute values? It is not clear to me how these data gaps were accounted for in the analysis.

Answer:

The thermal sensor in the camera is very sensitive to external heating. Specifically one-sided solar heating creates a thermal imbalance from one side of the image to the other. The resetting mechanism ensures that the sensor can adjust to this imbalance of external heating. Therefore, this resetting mechanism is necessary otherwise the thermal imbalance of the image would increase over time.

When the data was resampled to 2 Hz, the two NA-frames caused by the resetting mechanism were replaced with the last available image. Before the calculation of the A-TIV the corresponding frames were removed from the perturbation time series to ensure for a continuous velocimetry estimation. This shortened the signal by 26 seconds and caused the shorter time coverage of the A-TIV signal visible in figure 9.

Reviewer 2:

P23 line 305: this sentence is not clear to me

Answer:

A-TIV estimates velocity based on changes in the spatiotemporal patterns of the brightness temperature. Surface brightness temperature changes in response to rapid heat exchange occurring between the surface and near-surface turbulent air that tends to have a coherency associated with it. However, we know from eddy covariance measurements that turbulent coherent structures are 3D structures and as a result the 2D thermal patterns detected by the a TIV method will be a convolution of these 3D

dynamics. Hence, A-TIV is not necessarily capturing the exact movement of the coherent structure due to the lack of any explicit vertical measurements via infrared.

Reviewer 2:

P23, line 312: If I am correct then EC measurements are missing mostly the lower frequencies (larger eddies). Can you put this sentence a bit more into context with your experiment?

Answer:

With this paragraph we wanted to point out the advantage of combining the traditional point measurement methods with the newly proposed A-TIV / infrared camera measurements. We have rephrased the paragraph accordingly (330 - 336)

Reviewer 2:

P23, line 323: A very general question: can you describe why one would expect that the air and surface temperature perturbations show similar magnitudes given differences in thermal properties?

Answer:

This depends entirely on the surface type and its water content. As seen in the wheat stubble experiment the canopy decreases the effect of the thermal interaction of atmosphere and surface. It is expected that short cut grass and turf react well and immediately to temperature changes by the atmosphere adjacent to the surface. Additionally, other factors such as cloud cover, surface and soil moisture creating latent heat play a major role in the surface temperature perturbation magnitudes.

Reviewer 2:

P24 line 344: It would be helpful if you could link these statements to the single figures that support these claims?

Answer:

We have linked each statement to the figures from the results section (line 424 - 432).

Reviewer 2:

P24 line 356: this sentence is a bit unclear to me

Answer:

We have added a subordinate clause to clarify the sentence (line 435)

---

## Author Response (AR2)

General Answer:

Thank you for your time and consideration to review the manuscript again. We are delighted to present to you the answers to the minor revisions. We have addressed all your concerns.

Reviewer 2:

The authors edited the manuscript and added new information.
However, in my opinion the comparison of A-TIV with the TIV algorithm is still not detailed enough. The authors added a table showing the percentage of vacant cells and the importance of single TIVs to the A-TIV velocity estimate. However, it remains unclear how the different velocity fields agree with the reference data. I encourage the authors to add a figure comparing wind speed and wind and temperature perturbations from TIV and A-TIV and add the numbers for TIV to Table 4.

Answer:

We have added the calculation and figures of the requested TURF-T1 comparison to the appendix (Section appendix B) and have noted the quantitative difference of wind speed, wind direction of A-TIV compared to TIV in Table 4.

Reviewer 2:

I think more discussion is needed on the poor estimate of the mean wind speed. While it would be beneficial to have a spatially distributed estimate of wind velocity e.g. for the estimation of ET using energy balance models, the absolute magnitude of wind speed is of course very important too. I find it interesting that in Figure 5 the histograms of the physical and virtual TC arrays agree very well, even though the TC array is mounted 1.5 m above the ground. In the text it is stated that the sonic anemometer is also mounted at 1.5 m height. Maybe the authors could add the comparison of the TC array and sonic anemometer to their scatter plot in Figure 11 and discuss the reasons for the differences in wind speed in a bit more detail.

Answer:

The reviewer has misunderstood the mounting height of the thermocouples. It is stated clearly in the manuscript that the TC array reflects the situation 1.5 centimetre not 1.5 metre above the ground.

Reviewer 2:

The authors state in the Discussion p. 25, 362 "However, the histograms show that the distributions are not comparable, which is expected comparing a point measurement to a spatial approach." The authors should also discuss this aspect a bit more.

Answer:

We have added the following sentences to the paragraph:

"However, the histograms show that the distributions are not comparable, which is expected comparing a point measurement to a spatial approach. This means that the A-TIV is reflecting a spatial measurement whereas the other measurement methods are based on single point measurements which depend on their mounting height respectively their footprint. The direct spatial measurement of A-TIV reflects the atmospheric situation directly adjacent to the surface and hence, when compared to point measurements further away from the surface, may not reflect the same conditions."

Reviewer 2:

Stull, 1988: "Instead of observing a large region of space at an instant in time, we find it easier to make measurements at one point in space over a long time period. …. Thus, the wind speed could be used to translate turbulence measurements as a function of time to their corresponding measurements in space." The authors should discuss why this assumption does not hold in this application.

Answer:

We have added the following paragraph to the discussion section:

"The assumption from Stull, 1988 that meteorologists rather observe atmospheric conditions over longer periods of time (> several hours), than creating short observations over a large region of interest does not reflect the strategy of A-TIV applications. According to Stull, 1988 the long term point measurements can be translated to their corresponding spatial measurements as a function of time. A-TIV is a new approach in the sense that the measurement type is directly spatial and hence short term observations can immediately reflect the spatial component of turbulence. Moreover, the new type of data that is retrieved needs new spatiotemporal statistics and new anaylsis methods such as A-TIV for new insights into spatial turbulence."

Some specific comments:

Reviewer 2:

P.4, line 113: Did the authors really want to refer to Section 2.5?

Answer:

We have corrected this to Section 3.2.1

Reviewer 2:

P. 5, line 120: Could the authors add the used interval settings for each flight somewhere in the text or e.g. Table 1?

Answer:

We have added the automatically selected interval settings to the table.

Reviewer 2:
P. 5, line 140: In my opinion, this sentence belongs to 2.1 since it gives information about parameters used in the TIV algorithm, which are not specific to the A-TIV algorithm.

Answer:

We have moved the following three sentences of the paragraph to the end of section 2.1.

"TIV used previously a correlation technique presented by \cite{Kaga1992} called the greyscale correlation technique which uses simple pixel by pixel subtraction to obtain a correlation value (Inagaki, 2013). The A-TIV is usually calculated using the same technique with a correlation window size of 16 x 16 pixels and a search area size of 32 x 32 pixels. These settings were previously investigated as the most accurate (Schumacher, 2019).

Reviewer 2:
P. 7, line 143: The height of the sonic anemometer should be added to Section 2.3.

Answer:

The height has been mentioned in Section 2.3 in the following sentence:

"TURF-T1 was equipped with one sonic anemometer and TURF-T2 was equipped with two sonic anemometers, one in the grass field and one in the turf (Figure~\ref{fig:experimentsites}). All anemometers were mounted at 1.5 m above ground level, sampled at 20 Hz, and were placed in the field of view of the camera. "

Reviewer 2:
P. 11, line 200: To which frequency were the other experiments subsampled?

Answer:

All experiments were subsampled in the same way.  We have adjusted the sentence to:

"To evaluate in a first step the brightness temperature data captured by the infrared camera with the TC derived air temperature, the same methodology was applied to a "virtual" array taken from the brightness temperature perturbations which was sub-sampled using mean-sampling to a sampling rate of 20 Hz."

Reviewer 2:
P. 12, line 209: I do not understand why the output frequency necessarily has to be 2 Hz due to the ten second windows in the time series.

Answer:

To retrieve lag-cross correlations of the Thermocouple signal it is necessary to retrieve chunks from the signal and correlate it. Considering 20 Hz thermal data and chunks of 10 seconds (200 datapoints) and no overlapping of the chunks, this would mean 60 measurements. However to be able to compare to A-TIV and it's subsample to 2 Hz data it is necessary to overlap the chunks to retrieve a comparable time series.

Reviewer 2:

P. 14, line 272: Reporting a p-value > 0.95 is not very common.

Answer:

This refers most likely to the answer of the major revision and not to the manuscript itself. Hence, we will take this into account for our future reporting and responses.

Reviewer 2:
P. 14, Figure 5: I again encourage the authors to add legends to their plots where needed.

Answer:

We have added a legend to the plot as per your suggestion.

Reviewer 2:
P. 18, line 281: Did the authors really want to refer to Table 1 here?
Answer:

Yes. This is to reference to the higher wind speed of the two TURF-T experiments.

Reviewer 2:
P. 18, Table 4: The mean TIV speed should be added here too.

Answer:

The TIV mean speed was added according to your above comment.

Reviewer 2:
P. 23, line 309: The wheat stubble alters the wind profile and this also affects the exchange of heat between the surface and the atmosphere.
Answer:

We have added the following sentence:

"Furthermore, the wheat stubble alters the wind profile above it more significantly compared to the smooth roughness elements of TURF-T1 and TURF-T2 which also affects the exchange of heat between the surface and the atmosphere. "

P. 25, line 377: This sentence is not clear to me.

Answer:

We have adjusted the sentence as follows:

"The display of very small velocities ($<$ 0.5 m/s ) is also not ideal due to the a high range of extracted velocities from the multiple TIVs neglecting the display of small velocities"

P. 26, line 378: I don't really see that Figure 11 supports this claim.

Answer:

We have removed the Figure from the claim.

*""*